# A two-stage framework for enhancing crsyptocurrency portfolio performance: Integrating credibilistic CVaR criterion with a novel asset preselection approach

Hossein Ghanbari[1], Sina Tavakoli[1], Mostafa Shabani[1], Emran Mohammadi[1]*, Seyed Jafar Sadjadi[1], Ronald Ravinesh Kumar[2]

**1** Department of Industrial Engineering, Iran University of Science and Technology, Tehran, Iran,
**2** Department of Economics and Finance, The Business School, RMIT University, Saigon South Campus, Ho Chi Minh City, Vietnam

* E_mohammadi@iust.ac.ir

## Abstract

In an increasingly diverse investment landscape, the cryptocurrency market has emerged as a compelling option, offering the potential for high returns, diversification opportunities, and significant liquidity. However, the inherent volatility and regulatory uncertainties of this market present substantial risks, underscoring the need for a well-structured investment strategy. Among the various strategies available, portfolio optimization has become a dynamic and evolving area of focus in finance. Despite advancements in financial modeling, traditional portfolio optimization models often fall short, as uncertainty remains a fundamental characteristic of capital markets. To address this challenge, this paper integrates credibility theory with the Conditional Value-at-Risk (CVaR) framework, harnessing their combined strengths in modeling downside risk and managing uncertainty. Nevertheless, relying solely on this model may not be sufficient for achieving optimal investment outcomes, as portfolio optimization models often neglect the crucial step of selecting high-quality assets. This highlights the essential need for a robust pre-selection process. To tackle this issue, this paper introduces a novel pre-selection framework based on Multi-Attribute Decision Making (MADM) methods. Acknowledging that different MADM approaches can yield varying results—which creates uncertainty regarding the most reliable method—this research proposes a systematic framework for asset evaluation. By considering these factors, this paper proposes a two-stage framework for enhancing cryptocurrency portfolio performance. Stage 1, involves establishing comprehensive performance criteria for cryptocurrencies and employing a novel method for asset pre-selection. Stage 2 focuses on optimizing the selected assets using a credibilistic CVaR model, while considering practical constraints from real-world investment scenarios. The results of this two-stage framework demonstrate its effectiveness in

**Data availability statement:** All relevant data are within the manuscript and its Supporting Information files.

**Funding:** The author(s) received no specific funding for this work.

**Competing interests:** The authors have declared that no competing interests exist.

constructing well-diversified and efficient portfolios, addressing both the challenges of asset pre-selection and the complexities associated with uncertainty. By integrating these methodologies, investors can navigate the risks associated with cryptocurrency investments more effectively while maximizing potential returns.

## 1. Introduction

Investment plays a crucial role in fostering economic growth while enhancing personal financial well-being. It enables individuals and institutions to grow their wealth over time, build financial security, and contribute to the overall prosperity of society. One of the primary advantages of investing is the ability to generate passive income. By investing in assets such as stocks, bonds, or real estate, individuals can earn dividends, interest, or rental income, supplementing their primary sources of income. This passive income can be reinvested to further increase wealth or used to fund their lifestyle, providing financial stability and independence [1]. Moreover, investing plays a crucial role in the economy by channeling funds toward productive activities, fostering innovation, creating jobs, and ultimately driving economic growth [2].

In recent years, the investment environment has expanded significantly, offering a wider range of market opportunities for investors [3]. Among these emerging markets, the cryptocurrency market has garnered significant attention and popularity, driven by its unique characteristics and potential for high returns [4]. Cryptocurrency offers several key benefits that make it an attractive investment option. First, it provides opportunities for diversification, allowing investors to reduce their exposure to traditional assets such as stocks and bonds [5]. Cryptocurrencies are also known for their high liquidity, enabling investors to buy and sell quickly at competitive prices. Additionally, the decentralized nature of cryptocurrencies offers greater transparency and security compared to traditional financial systems, as blockchain technology ensures that transactions are recorded in a secure, immutable ledger [6]. Moreover, cryptocurrencies have the potential for high growth due to their evolving use cases, including decentralized finance, digital assets, and smart contracts, making them appealing to risk-tolerant investors seeking substantial returns [7,8]. However, while this market presents significant investment opportunities, it also carries substantial risks due to its inherent volatility, regulatory uncertainties, and market dynamics [9,10]. Price fluctuations in the cryptocurrency market can be sudden and extreme, posing challenges even to experienced investors [11]. Therefore, implementing a well-structured investment strategy is essential for managing risks, preserving capital, and maximizing long-term returns. A wide range of investment strategies has been developed, each designed to align with specific risk tolerances and financial objectives. Among these approaches, portfolio optimization has emerged as one of the most effective and widely applied methods in modern investment management.

Portfolio optimization is a dynamic and evolving topic in the fields of finance and investment, playing an important role in modern asset management [12,13]. The concept was first introduced by Harry Markowitz [14] in 1952 through his groundbreaking

work on Modern Portfolio Theory (MPT), which laid the foundation for systematic investment strategies. A portfolio, in the context of investment, refers to a collection of financial assets such as stocks, bonds, commodities, real estate, or cryptocurrencies held by an individual or institution. The primary goal of building a portfolio is to balance risk and return by diversifying investments across different asset classes. Diversification reduces the overall risk of the portfolio because the performance of various assets may not be perfectly correlated; when some assets decline in value, others may perform well, offsetting potential losses. Markowitz's portfolio theory introduced the concept of efficient portfolios, which offer the highest expected return for a given level of risk or the lowest risk for a desired level of return. His mean-variance optimization framework uses expected returns, variances, and covariances of asset returns to determine optimal asset allocation. This approach remains a cornerstone of investment management, guiding both individual and institutional investors in constructing portfolios that align with their financial goals and risk tolerance. Portfolio optimization has attracted significant attention from both investors and researchers, driving continuous advancements in investment management techniques. As the financial markets evolved, the need for more robust risk management tools became evident, prompting researchers to develop a variety of risk measures tailored to different investment environments. Among these measures, Value at Risk (VaR) [15] and Conditional Value at Risk (CVaR) [16,17] have gained considerable prominence [18]. Both belong to the class of downside risk measures, focusing on potential losses rather than overall variability. VaR estimates the maximum expected loss over a specific time frame at a given confidence level, while CVaR goes a step further by assessing the average loss beyond the VaR threshold, providing a more comprehensive view of extreme risks [19,20]. Due to the highly volatile and unpredictable nature of the cryptocurrency market, downside risk measures like VaR and CVaR have better applicability in this context. They enable investors to evaluate and manage extreme losses, making them valuable tools for constructing more resilient and risk-aware investment portfolios in the crypto market.

However, despite advances in financial modeling, uncertainty remains a fundamental characteristic of capital markets, as much of the information used in investment decision-making is inherently uncertain, imprecise, or incomplete [21]. To tackle the challenges posed by this uncertainty, researchers have explored alternative approaches such as fuzzy set theory, first introduced by Zadeh [22]. Building on this concept, Liu [23] proposed credibility theory, which was further expanded in subsequent works [24]. Credibility theory offers several advantages in addressing uncertainty within financial markets. Unlike traditional probabilistic models that rely on precise statistical distributions, credibility theory provides a flexible framework for handling imprecise and incomplete data. It combines elements of probability and possibility theory, making it well-suited for environments characterized by ambiguity and vagueness. One key benefit of credibility theory is its ability to model uncertain asset returns using a more realistic representation of market conditions. This approach accounts for both optimistic and pessimistic scenarios, enabling more comprehensive risk assessments. Additionally, credibility measures are computationally efficient and can be easily integrated into portfolio optimization models, enhancing their practical applicability. Credibility theory has been applied in several portfolio optimization studies, demonstrating its effectiveness in managing uncertainty and enhancing investment strategies. However, its application in cryptocurrency portfolio optimization remains relatively underexplored. Given the unique volatility and uncertainty of the crypto market, further investigation is needed to harness the potential of this theory in this emerging asset class.

To address this research gap, this paper integrates credibility theory with the CVaR framework, leveraging their combined strengths in modeling downside risk and managing uncertainty. Additionally, practical constraints commonly encountered in real-world investment scenarios are considered, ensuring that the proposed framework is both theoretically sound and practically applicable in the dynamic cryptocurrency market. However, relying solely on portfolio optimization models may not be sufficient to achieve optimal investment outcomes. While these models offer valuable frameworks for asset allocation, they often overlook the critical step of selecting high-quality assets. A thorough pre-selection process is essential to identify assets with strong fundamentals, growth potential, and resilience to market fluctuations. By focusing on high-potential assets before applying portfolio optimization models, investors can enhance the effectiveness of their strategies. This ensures that the portfolio includes assets well-positioned for growth while minimizing potential risks. In the

cryptocurrency market, pre-selection is particularly important due to the presence of numerous assets with limited investment value. Various methodologies have been proposed for asset pre-selection, including Data Envelopment Analysis (DEA), machine learning, and deep learning techniques. Among these, Multi-Attribute Decision-Making (MADM) methods have proven effective because they can evaluate assets based on multiple criteria, which is crucial for investors making informed decisions. However, a notable challenge with MADM methods is that different methods can produce varying results, creating uncertainty about which approach provides the most reliable outcome. To address this issue, this paper proposes a novel pre-selection framework based on MADM methods that delivers robust and consistent results. To the best of our knowledge, no existing study has applied a systematic pre-selection process in the context of cryptocurrency portfolio optimization. Therefore, this paper first introduces a comprehensive set of cryptocurrency performance criteria and utilizes the proposed pre-selection method based on these criteria. By integrating these components, this paper presents a two-stage framework for enhancing cryptocurrency portfolio performance. In the first stage, after defining relevant cryptocurrency performance criteria, we apply the novel pre-selection method. In the second stage, we implement a credibilistic CVaR model with practical investment constraints, creating a comprehensive and effective investment strategy for the cryptocurrency market.

The remainder of this paper is organized as follows: Section 2 provides a comprehensive and systematic literature review in four subsections, examining prior research on (I) cryptocurrency portfolio optimization, (II) portfolio optimization with asset pre-selection, (III) the application of credibility theory in portfolio optimization, and (IV) the identified research gap. Section 3 outlines the research methodology, beginning with the introduction of key concepts and definitions related to each stage of the proposed framework. This is followed by introducing alternatives and defining cryptocurrency performance criteria essential for the asset pre-selection process in Stage 1. Finally, the proposed optimization model for Stage 2, based on the credibilistic CVaR approach with practical investment constraints, is developed in detail. Section 4 presents the empirical results and computational analysis of the proposed two-stage framework, applying real-world cryptocurrency market data to evaluate its performance. Section 5 provides an in-depth discussion, critically analyzing the theoretical foundations and empirical outcomes of the proposed framework. Section 6 concludes the study by summarizing key findings and suggesting potential directions for future research.

## 2. Literature review

This section provides a comprehensive review of the relevant literature, divided into four subsections to cover all key aspects of the research. The first subsection reviews existing studies on cryptocurrency portfolio optimization, highlighting various methods and frameworks used in this emerging market. The second subsection focuses on portfolio optimization with asset pre-selection, emphasizing techniques employed to enhance investment outcomes through prior asset filtering. The third subsection explores the application of credibility theory in portfolio optimization, discussing its role in managing uncertainty and improving risk assessment. Finally, the fourth subsection identifies the research gap, outlining unexplored areas and justifying the need for the proposed two-stage framework.

### 2.1. Cryptocurrency portfolio optimization

Cryptocurrency portfolio optimization has become a prominent area of research, fueled by the inherent complexity and volatility of digital asset markets. As the cryptocurrency sector continues to expand, researchers and financial analysts have concentrated on crafting innovative models and strategies to effectively balance risk and maximize returns in this dynamic investment landscape. This literature review examines the wide range of methodologies and techniques applied to cryptocurrency portfolio optimization, highlighting significant findings and advancements that contribute to shaping the development of this fast-evolving domain.

James and Menzies [25] investigated whether the cryptocurrency market exhibits mathematical properties comparable to those of the equity market. Departing from traditional portfolio theory, which is grounded in the financial behavior of

equity securities, their research focused on the purchasing patterns of retail cryptocurrency investors. The study emphasized collective market dynamics and portfolio diversification within the cryptocurrency domain, exploring the applicability of equity market findings to digital assets. Bowala and Singh [26] developed a data-driven risk forecasting approach tailored to cryptocurrency portfolios, addressing the skewness and kurtosis of returns. Critiquing traditional risk measures for their normality assumptions, they utilized high-frequency data to better capture volatility dynamics. Results showed superior performance in optimizing cryptocurrency portfolios, providing a robust framework for risk management in this volatile asset class. Sahu et al. [27] compared portfolio optimization methods and short-term strategies for the cryptocurrency market using high-frequency data from the top ten cryptocurrencies by market capitalization. The study evaluated Sharpe ratio maximization and kurtosis minimization to balance returns and risks, offering insights into optimizing portfolios in dynamic market conditions. Chen [28] examined the relevance and effectiveness of modern portfolio theory (MPT) in portfolios that include cryptocurrencies. The study assessed whether traditional MPT principles, such as diversification and risk–return optimization, remain applicable in the context of the high volatility and unique risk-return profiles characteristic of cryptocurrencies. Jeleskovic et al. [29] explored the potential benefits of integrating cryptocurrencies into traditional investment portfolios. Using a GARCH-Copula model within the Markowitz framework, the study evaluated whether such integration enhances portfolio performance, particularly by improving the Sharpe ratio and overall portfolio stability, offering new perspectives on optimizing mixed-asset portfolios. Kim et al. [30] examined the risk–return profiles of traditional, cryptocurrency, and hybrid portfolios, focusing on the impact of cryptocurrency integration into global portfolios. Using ensemble methods and tracing strategies, the study analyzed allocation ratios of 1%, 3%, and 5% across optimization techniques, including minimum variance, maximum diversification, equal risk contribution, and hierarchical risk parity. Results highlighted the influence of allocation levels on returns, volatility, Sharpe ratios, and maximum drawdowns, providing critical insights for optimizing portfolios with cryptocurrency exposure. Hrytsiuk et al. [31] introduced a modified Markowitz model for cryptocurrency portfolios, substituting the variance-based risk metric with VaR. By incorporating the Cauchy distribution characteristic of cryptocurrency returns, this approach addressed the standard model's reliance on normality assumptions. The findings highlighted improved risk assessment and portfolio robustness, effectively capturing the heavy tails and extreme risks of volatile digital asset markets. Brauneis and Mestel [32] applied the Markowitz mean-variance framework to assess diversification benefits in cryptocurrency portfolios using daily data from the top 500 cryptocurrencies over three years. The study compared naïve diversification with optimization strategies targeting maximum return and minimum variance. Results demonstrated that diversified portfolios significantly reduced risk and outperformed single-asset investments, such as holding only Bitcoin, in overall performance. Ma et al. [33] used data from 2015 to 2019 to analyze the impact of integrating Bitcoin (BTC), Ethereum (ETH), Ripple (XRP), Bitcoin Cash (BTC), and Litecoin (LTC) into traditional portfolios. The study evaluated diversification effects and risk-return improvements across asset classes using various optimization techniques, emphasizing Ethereum and Bitcoin's superior diversification benefits. Mba et al. [34] proposed two advanced cryptocurrency portfolio optimization models, GARCH-DE and GARCH-DE-t-copula, comparing them to the traditional Differential Evolution (DE) model. These models, tested in single- and multi-period frameworks, addressed complex dependency structures and extreme risks in cryptocurrency markets. The GARCH-DE-t-copula model, incorporating a t-copula, effectively captured tail dependencies and volatility clustering, demonstrating superior risk management and return optimization under market volatility. Aljinović et al. [35] introduced a multicriteria portfolio optimization approach leveraging the PROMETHEE II method, expanding beyond traditional return and risk measures. Their model incorporated diverse criteria, such as market capitalization, trading volume, VaR, CVaR, and the overall attractiveness of cryptocurrencies. Using data from January 2017 to February 2020, they demonstrated that their multicriteria approach achieved superior out-of-sample portfolio performance compared to conventional optimization models across various risk and return metrics. Maghsoodi [36] proposed a hybrid decision support system for cryptocurrency portfolio management, integrating time series forecasting via the Prophet Forecasting Model (PFM) with the enhanced CLUS-MCDA II algorithm. This system utilized advanced clustering methods, such as DBSCAN, alongside

multicriteria decision analysis techniques like VIKOR and MULTIMOORA, to optimize allocation across more than 70 cryptocurrencies. Their model provided investors with a robust and informed tool to navigate the highly dynamic cryptocurrency market. Mba and Mwambi [37] developed the Markov-switching COGARCH-R-vine (MSCOGARCH) model to optimize cryptocurrency portfolios by addressing structural breaks, heavy tails, and volatility clustering. Compared to a single-regime COGARCH model, the MSCOGARCH model demonstrated superior risk estimation and portfolio optimization by accommodating regime changes in volatility. This approach offered enhanced flexibility for managing portfolios in the volatile cryptocurrency market. Ali et al. [19] explored the diversification benefits of green cryptocurrencies in the context of global portfolio optimization, introducing a novel four-step process to identify cryptocurrencies that are more energy-efficient than others. This study highlights the growing concern over the environmental impact of cryptocurrencies, particularly energy-intensive mining practices. By focusing on energy-efficient cryptocurrencies such as Cardano (ADA), Tezos (XTZ), and Stellar (XLM), the authors demonstrate how these green cryptocurrencies can serve as effective diversifiers for portfolios, offering benefits comparable to or even superior to their non-green counterparts like Bitcoin (BTC) and Ethereum (ETH). The research employed a variety of advanced econometric models, including dynamic conditional correlation (DCC-GARCH) and four-moment modified VaR, to evaluate the downside risk and expected shortfall of portfolios containing both green and non-green cryptocurrencies. In a recent study, Ghanbari et al. [38] proposed a robust framework for cryptocurrency portfolio optimization, leveraging the credibilistic CVaR criterion to address the distinct characteristics of digital asset markets. Recognizing the high volatility, frequent price swings, and the inherent uncertainty of cryptocurrencies, their approach models asset returns using fuzzy logic, enabling greater accuracy in risk assessment. Their proposed framework accounts for the unpredictable behavior of the crypto market, including extreme tail risks, offering a tailored solution for managing digital asset portfolios. This advancement underscores the potential of credibility-based models in navigating the complexities of cryptocurrency investment.

## 2.2. Portfolio optimization with asset preselection

Pre-selection plays a vital role in portfolio management by identifying and selecting assets prior to the portfolio optimization process. This step is particularly critical in volatile markets such as cryptocurrencies, where asset selection significantly affects portfolio performance. Pre-selection involves various methodologies aimed at filtering and selecting high-potential assets. This literature review explores the significance of pre-selection in portfolio optimization, emphasizing key developments and notable findings that shape this rapidly evolving field.

Lozza et al. [39] conducted an ex-post analysis of asset preselection frameworks, employing the joint Markovian dynamics of asset returns within stochastic market boundaries. Examining approximately 10,000 equities across 14 international markets, their findings substantiated the superior efficacy of Markovian-based methodologies over traditional Sharpe ratio optimization, emphasizing the strategic significance of probabilistic state transitions in enhancing portfolio efficiency amidst intricate market dynamics. Huang [40] developed an advanced stock selection paradigm integrating Support Vector Regression (SVR) for performance forecasting and Genetic Algorithms (GA) for parameter optimization and feature refinement. Equally weighted portfolios, derived from performance-ranked stocks, demonstrated empirically superior returns, affirming the model's efficacy over conventional benchmarks in optimizing investment outcomes. Nguyen [41] introduced a sophisticated risk-measurement framework for large-scale datasets, integrating a stock preselection mechanism to exclude low-diversification stocks pre-optimization. By leveraging performance metrics such as the Sharpe ratio, Stutzer index, and Omega measure, the methodology enhanced portfolio construction. Empirical findings confirmed that preselection markedly improved portfolio performance and diversification, addressing critical challenges in large-scale optimization. Rather et al. [42] introduced a robust hybrid model for stock return prediction, integrating linear models, specifically the Autoregressive Moving Average and Exponential Smoothing techniques, with the nonlinear capabilities of a Recurrent Neural Network (RNN). By combining these methods, the framework leveraged their complementary strengths, while GA optimized the model's weight distribution, ensuring balanced contributions. Experimental results demonstrated

the hybrid model's significant advantage over standalone RNNs, achieving superior predictive accuracy. Le Caillec et al. [43] proposed a stock selection model integrating behavioral uncertainty and probabilistic techniques, utilizing Cumulative Return and multiple Technical Indicators for preselection. Empirical analysis affirmed its efficacy in enhancing portfolio performance, addressing traditional strategy limitations by incorporating both technical analysis and behavioral dynamics. Fischer and Krauss [44] utilized a LSTM neural network to predict S&P 500 stock movements (1992–2015), outperforming models without memory functions (e.g., RF, DNN, LR). LSTM-based portfolios consistently exceeded alternatives, affirming the superiority of memory-enhanced architectures in financial time series analysis. Alizadeh et al. [45] developed a portfolio optimization model combining an adaptive neural fuzzy inference system for return prediction with a variance index for risk assessment. The model outperformed traditional Mean-Variance, neural network, and Sugeno–Yasukawa approaches, demonstrating the efficacy of integrating AI techniques with modern portfolio optimization for enhanced investment performance. Paiva et al. [46] introduced a unified decision-making model for day-trading in stock market investments, combining Support Vector Machines (SVM) with the MV framework for portfolio selection. The model was benchmarked against two alternatives: SVM + 1/N and Random+MV. Experimental results using assets from the Ibovespa stock market demonstrated that the proposed model delivered superior performance, highlighting its effectiveness in day-trading scenarios. Wang et al. [47] proposed a portfolio construction approach combining LSTM networks with the MV model. The LSTM network was employed to identify stock price patterns using various technical indicators, including the Relative Strength Index (RSI), Momentum Index (MOM), and True Range (TR). The MV model then optimized portfolios composed of five to ten assets. Comparative experiments revealed that the LSTM+MV method consistently outperformed other machine learning and MV-based models, particularly when portfolios included ten stocks. Ta et al. [48] constructed a portfolio using a LSTM neural network alongside three portfolio optimization techniques: the equal weight method, Monte Carlo simulation, and the Mean-Variance (MV) model. For comparison, linear regression and SVM were also applied in the stock selection process. Test results demonstrated that the LSTM neural network surpassed linear regression and SVM in prediction accuracy, and the portfolios it generated outperformed those built using the alternative methods. Chen et al. [49] proposed an innovative portfolio construction method integrating eXtreme Gradient Boosting (XGBoost) with an improved firefly algorithm (IFA) for stock price prediction. The Mean-Variance (MV) model was subsequently used to select and optimize portfolios containing varying numbers of stocks, focusing on those with higher predicted returns. Empirical evaluations revealed that the proposed hybrid model outperformed benchmark models, with its effectiveness being particularly notable when the portfolio consisted of seven stocks.

## 2.3. Portfolio optimization using credibility theory

Credibilistic portfolio optimization has gained significant attention in research due to its effectiveness in addressing uncertainty and ambiguity common in financial markets. Grounded in credibility theory and supported by fuzzy logic, this approach provides a robust framework for optimizing portfolios under conditions of high market volatility. Its capacity to incorporate expert opinions and subjective assessments into the investment decision-making process enhances its relevance in complex financial environments. This literature review explores the application of credibility theory in portfolio optimization, highlighting key methodologies, notable advancements, and empirical findings that underscore its value and potential in contemporary portfolio management.

Liu et al. [50] developed a credibilistic CVaR-based portfolio optimization model, enhancing traditional mean-variance frameworks by integrating CVaR of fuzzy variables to distinguish downside risks from upside potential. Solved via deterministic mixed-integer programming, the model ensures computational efficiency and refined risk management in portfolio optimization. Mohebbi and Najafi [51] proposed a multi-period fuzzy portfolio optimization model integrating credibility theory and scenario tree analysis to address market uncertainty. Using a bi-objective VaR framework, the model optimizes portfolios while incorporating transaction costs, risk-free investments, and practical constraints such as cardinality, thresholds, classes, and liquidity. Solved via interactive dynamic programming, the model combines fuzzy set theory with

scenario analysis, offering a robust, adaptive tool for balanced portfolio optimization under real-world conditions. Deng et al. [52] developed a fuzzy mean-entropy portfolio optimization model leveraging credibility theory to enhance risk measurement and portfolio selection under uncertainty. The model integrates entropy as a risk measure, arguing its superiority over variance, especially in the context of fuzzy financial markets. It also incorporates transaction costs, addressing practical considerations often overlooked in traditional models. A sensitivity analysis was performed to assess the influence of parameter variations on the optimal portfolio configuration. The proposed approach aims to equip investors with a reliable and stable tool for portfolio optimization in uncertain and transaction-cost-sensitive financial environments. Liu et al. [53] proposed a multi-period portfolio optimization model designed to incorporate bankruptcy control within a fuzzy economic framework using credibility theory. The model seeks to maximize terminal wealth while simultaneously minimizing cumulative risk and uncertainty throughout the investment horizon. To enhance decision-making, it integrates affine recourse, addressing the impact of historical prediction biases on current portfolio adjustments. The authors also developed a hybrid particle swarm optimization algorithm to solve the model efficiently, providing a practical and robust tool for investors to manage risk and prevent bankruptcy in complex multi-period investment scenarios. Gupta et al. [54] introduced a multi-period portfolio optimization model utilizing coherent fuzzy numbers within a credibilistic framework to enhance decision-making under uncertainty. The model allows for greater flexibility in defining investor risk tolerance by incorporating mean absolute semi-deviation and CVaR as risk measures. It also addresses practical constraints, including cardinality, skewness, and transaction costs, to create realistic and adaptable investment strategies across multiple time horizons. The effectiveness of the proposed model was demonstrated through real-world case studies involving assets from the National Stock Exchange of India and major U.S. stock indices, noting its practical applicability and robustness in diverse market conditions. Mehlawat et al. [55] proposed a multiobjective portfolio optimization model employing coherent fuzzy numbers within a credibilistic framework. This model innovatively integrates investor attitudes—pessimistic, optimistic, or neutral—toward financial markets through a novel credibility function. It replaces variance with mean-absolute semi-deviation for a more realistic risk measure and incorporates skewness to account for the asymmetry of returns. Numerical examples and a genetic algorithm-based solution method highlight the model's ability to capture investor preferences and handle market uncertainties effectively, offering enhanced flexibility and precision compared to traditional approaches. García et al. [56] extended the traditional mean-semivariance portfolio selection model by developing a multiobjective credibilistic framework that includes the Price-to-Earnings Ratio (PER) as an additional performance criterion. Using L-R power fuzzy numbers to represent uncertainty in asset returns and PER, the model addresses limitations of the classical mean-variance framework while integrating real-world constraints such as budget, bounds, and cardinality. Empirical tests on stocks from the Latin American Integrated Market demonstrated the model's ability to generate a well-diversified set of efficient portfolios tailored to multiple objectives. In another study, García et al. [57] introduced an advanced multiobjective portfolio optimization model extending the stochastic mean-variance approach by incorporating fuzzy multiobjective criteria. Using trapezoidal fuzzy numbers, the model balances objectives related to return, risk, and liquidity, providing a robust framework for portfolio selection under uncertain and dynamic market conditions. Recently, Ghanbari et al. [38] extended the theoretical application of credibility theory in cryptocurrency portfolio optimization by incorporating trapezoidal fuzzy variables and the credibilistic CVaR framework to effectively model and quantify extreme tail risks. The inclusion of cardinality and allocation constraints further enhanced the model's robustness, offering a comprehensive approach to managing the stochastic complexities of digital asset markets. This work represents a significant contribution to the field of quantitative financial optimization, setting a valuable benchmark for future research in cryptocurrency investment strategies.

## 2.4. Research gap

After conducting a comprehensive literature review from various perspectives relevant to this study, several critical research gaps have been identified that this paper aims to address:

1.  *Lack of asset preselection in cryptocurrency portfolio optimization*

To the best of our knowledge, no existing study on cryptocurrency portfolio optimization incorporates an asset preselection stage, despite its crucial importance in such a highly volatile and diverse market. Given the wide range of cryptocurrencies with varying investment potentials, preselection is essential for filtering out low-quality assets. This paper addresses this gap by proposing a two-stage investment framework specifically designed for the cryptocurrency market. The first stage focuses on asset preselection, while the second stage involves portfolio optimization.

2.  *Absence of cryptocurrency performance criteria*

Due to the lack of studies involving asset preselection in cryptocurrency portfolio optimization, there is also no established set of performance criteria tailored for evaluating cryptocurrencies. This gap leaves investors without clear guidelines for assessing asset quality before portfolio construction. To address this gap, this paper proposes a comprehensive set of cryptocurrency performance criteria that reflect fundamental, technical, and market-driven factors relevant to digital asset evaluation.

3.  *Limitations of existing preselection methods*

Although various methodologies for asset preselection exist, MADM methods have emerged as one of the more effective approaches due to their ability to consider multiple evaluation criteria. However, a key limitation is that different MADM methods often produce varying results, causing confusion for investors when selecting assets. To address this issue, this paper proposes a novel preselection strategy based on MADM methods that generates robust and reliable results, reducing ambiguity in investment decision-making.

4.  *Neglect of market uncertainty in cryptocurrency portfolio optimization*

Uncertainty is an inherent characteristic of the cryptocurrency market due to its extreme volatility, limited historical data, and unpredictable market dynamics. Despite its importance, uncertainty has been insufficiently addressed in existing cryptocurrency portfolio optimization models. To fill this gap, this paper applies a credibilistic CVaR model, which effectively accounts for uncertainty by incorporating fuzzy logic principles, providing a more realistic and adaptive risk management approach. This model also considers practical constraints, enhancing its applicability and relevance in real-world investment scenarios.

## 3. Research methodology

This paper presents a two-stage framework for enhancing cryptocurrency portfolio performance, designed to address portfolio construction challenges in volatile and uncertain cryptocurrency markets. The framework consists of two stages: In stage 1, we perform pre-selection of high-potential assets by a novel asset preselection approach, and in stage 2, we optimize the selected assets using a credibilistic CVaR approach (see Fig 1).

This section outlines the research methodology in detail and is divided into two subsections. Section 3.1, introduces the pre-selection process of Stage 1, and Section 3.2, provides a detailed description of the optimization process in Stage 2.

### 3.1. Stage 1 - Asset preselection

In Stage 1, we focus on the pre-selection of high-potential assets using a novel approach grounded in MADM methods. A notable challenge with MADM techniques is that different methods can yield varying results, leading to uncertainty about which approach provides the most reliable outcome. To address this issue, this paper proposes a robust pre-selection framework designed to provide consistent outcomes. This framework first calculates the results using a variety

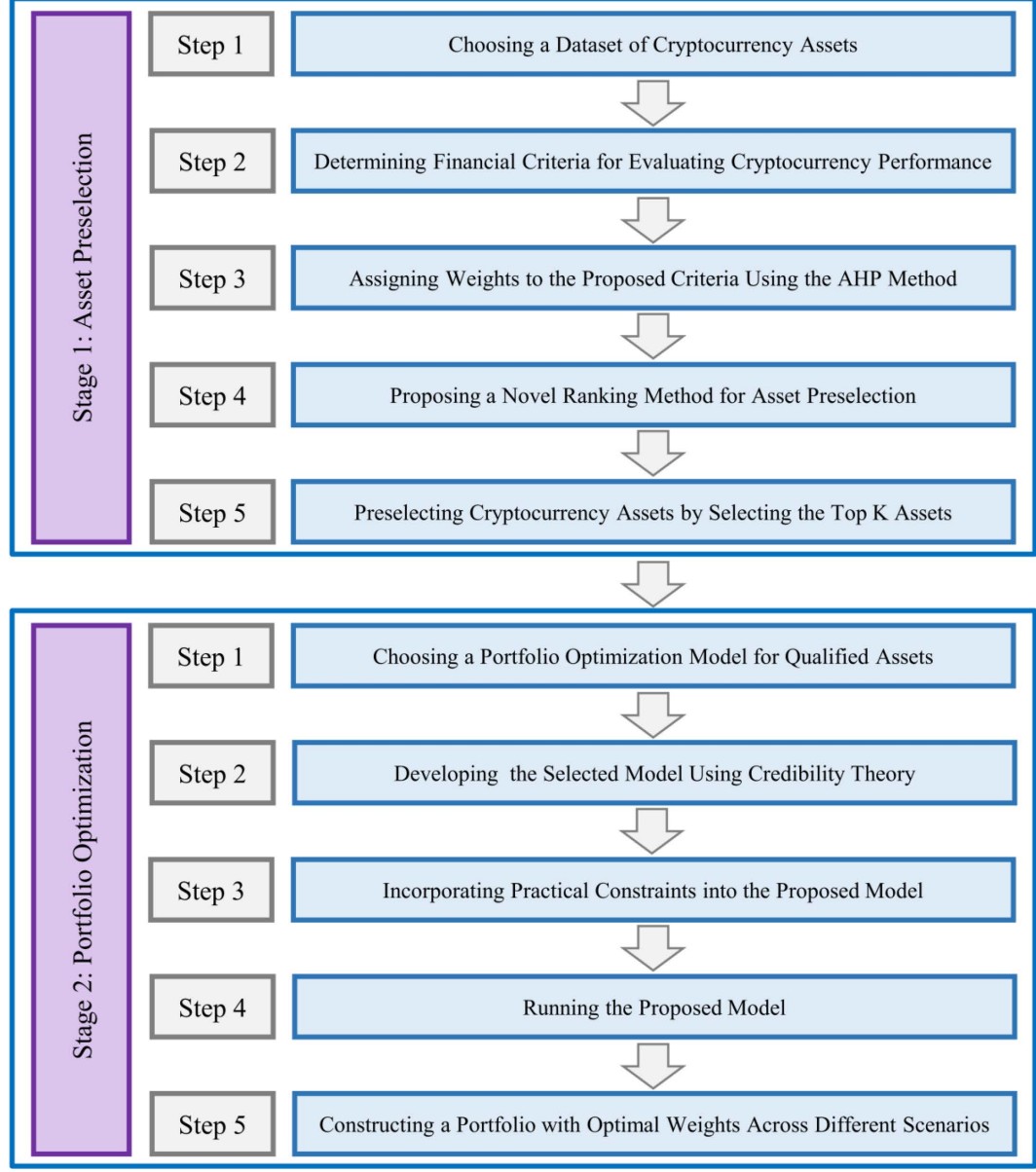

**Fig 1. The methodology of proposed two-stage framework for enhancing cryptocurrency portfolio performance.**

of methods, specifically employing 13 MADM techniques in this paper. Each of these methods offers unique strengths and perspectives in evaluating potential cryptocurrency assets. By utilizing multiple approaches, we aim to capture a comprehensive view of asset performance. Once the results are generated from these methods, they are systematically combined using the Copeland approach. This method evaluates and aggregates the outcomes, allowing us to rank the assets based on their overall performance across all selected MADM methods. By applying the Copeland approach, we enhance the reliability of the final selections, mitigating the inconsistencies that can arise from relying on a single MADM method. This dual-layered process not only strengthens our pre-selection framework but also ensures that the selected assets are those most likely to perform well in the dynamic cryptocurrency market.

In this section, we will first provide background knowledge on the 13 MADM methods and other approaches utilized in Stage 1 of our framework. Following that, we will describe the cryptocurrency dataset, which represents the alternatives in our analysis. In the third part, we will introduce the criteria for evaluating cryptocurrency performance. To the best of our knowledge, no existing study has implemented a systematic pre-selection process in the context of cryptocurrency portfolio optimization, resulting in a lack of benchmarks for these criteria. Therefore, we aim to establish a robust set of criteria for evaluating cryptocurrency performance, which can serve as a valuable benchmark for assessing cryptocurrencies during pre-selection or selection. Finally, we will employ the Analytic Hierarchy Process (AHP) method to determine the weight of each criterion, ensuring a systematic and rigorous evaluation framework.

**3.1.1. Background knowledge on MADM methods.** This paper employs a total of 13 MADM methods (see Table 1) to enhance the pre-selection process of cryptocurrency assets. These methods include MARCOS [58] (see Appendix 1 in S1 Appendix), CODAS [59] (Appendix 2 in S1 Appendix), CoCoSo [60] (Appendix 3 in S1 Appendix), EDAS [61] (Appendix 4 in S1 Appendix), WASPAS [62] (Appendix 5 in S1 Appendix), TOPSIS [63] (Appendix 6), MOORA [64] (Appendix 7), COPRAS [65] (Appendix 8), ARAS [66] (Appendix 9 in S1 Appendix), VIKOR [67] (Appendix 10 in S1 Appendix), MABAC [68] (Appendix 11 in S1 Appendix), MACBETH [69] (Appendix 12 in S1 Appendix), and TODIM [70] (Appendix 13 in S1 Appendix). Each method offers unique strengths and insights, collectively providing a robust framework for evaluating the potential of cryptocurrency assets in the context of our systematic pre-selection approach.

**3.1.2. Mean rank method.** The mean rank method is an effective decision-making technique used to evaluate multiple alternatives by assigning ranks based on specific criteria or performance metrics. Each alternative is assessed individually, receiving a rank relative to others—where the best-performing alternative is ranked 1, the second-best is ranked 2, and so on. After all ranks are assigned, the mean rank method calculates the average rank for each alternative. The alternative with the lowest average rank is then selected as the optimal choice, as a lower rank indicates better overall performance. This method is widely applicable in fields like finance, marketing, and operations management, offering a straightforward approach to streamline evaluations and minimize subjectivity in the ranking process.

**Table 1. Summary of employed MADM methods and corresponding references.**

| Method | Abbreviation | Reference |
|---|---|---|
| Measurement of Alternatives and Ranking according to COmpromise Solution | MARCOS | [58] |
| Combinative Distance-based Assessment | CODAS | [59] |
| COmbined COmpromise SOlution | CoCoSo | [60] |
| Evaluation based on Distance from Average Solution | EDAS | [61] |
| Weighted Aggregates Sum Product Assessment | WASPAS | [62] |
| Technique for Order of Preference by Similarity to Ideal Solution | TOPSIS | [63] |
| Multi-Objective Optimization on the basis of Ratio Analysis | MOORA | [64] |
| Complex PRoportional Assessment | COPRAS | [65] |
| Additive Ratio ASsessment | ARAS | [66] |
| VIseKriterijumska Optimizacija I Kompromisno Resenje | VIKOR | [67] |
| Multi-Attributive Border Approximation area Comparison | MABAC | [68] |
| Measuring Attractiveness by a Categorical Based Evaluation TecHnique | MACBETH | [69] |
| TOmada de Decisao Interativa e Multicriterio - Interactive and Multicriteria Decision Making | TODIM | [70] |

Source: Authors' own compilation

**3.1.3. Borda count method.** The Borda count method [71] is a systematic approach for ranking alternatives based on their performance in pairwise comparisons. It begins by constructing an $m \times m$ matrix, where $m$ represents the number of alternatives under consideration. Each entry in this square matrix is populated based on the number of wins each alternative achieves when compared to others. In this matrix, if an alternative in a given row has more wins than an alternative in a corresponding column, an "M" is placed in that entry. This notation signifies that the alternative in the row ranks higher than the one in the column across various decision-making scenarios. Conversely, if the number of wins in the row is equal to or less than that in the column, an "X" is recorded in that entry, indicating that the row's alternative ranks equally or lower than the column's alternative. After populating the matrix, the total number of wins for each alternative is calculated by summing the "M" entries in each row. This tally reflects the number of times each alternative has outperformed others in head-to-head comparisons. Finally, the alternatives are ranked based on the total number of wins, with those achieving a higher win count receiving a superior rank. This method not only provides a clear ranking but also emphasizes the comparative strengths of each alternative, making it a valuable tool in decision-making processes where multiple options need to be evaluated against one another.

**3.1.4. Copeland method.** Similar to the Borda count method, the Copeland method [72] also employs an $m \times m$ matrix to facilitate the ranking of alternatives based on their performance in pairwise comparisons. In this matrix, each entry is determined by the number of wins each alternative accumulates against others. Specifically, if an alternative in a given row has more wins than the alternative in a corresponding column, an "M" is placed in that entry. This indicates that the alternative in the row holds a higher rank in the context of various decision-making scenarios. Conversely, if the number of wins in the row is equal to or less than that in the column, an "X" is recorded, suggesting that the row's alternative ranks equally or lower than the column's alternative. Once the matrix is populated, the next step involves calculating the total number of wins for each alternative by summing the "M" entries in each row. This provides a clear indication of how many alternatives each option has outperformed. Additionally, the method also requires determining the number of losses for each alternative, which is done by summing the "M" entries in each column. The final ranking of the alternatives is derived from the difference between the total number of wins and losses for each option. Alternatives that exhibit a greater positive difference between their wins and losses are assigned higher ranks, reflecting their overall superiority in the comparison process. This approach not only allows for a nuanced ranking of alternatives but also emphasizes the relative strengths and weaknesses of each option, making the Copeland method a robust tool for decision-making in complex scenarios where multiple alternatives must be evaluated. By considering both wins and losses, the Copeland method provides a more balanced assessment compared to methods that focus solely on wins.

**3.1.5. Dataset description.** This section is dedicated to providing a comprehensive description of the dataset utilized in this study. The analysis focuses on 47 alternative cryptocurrencies, with data meticulously gathered from CoinGecko.com, a reputable platform known for its extensive cryptocurrency market data. The dataset encompasses daily price information spanning from December 1, 2023, to December 14, 2024. This timeframe allows for a thorough examination of price movements and trends in the rapidly evolving cryptocurrency market. From the daily price data, returns for each asset were calculated to assess their performance over the specified period. Descriptive statistics for the selected assets are summarized in Table 2, offering a clear view of key metrics such as mean, median, standard deviation, and other relevant statistical measures. This table serves as a crucial reference for understanding the characteristics of the assets under consideration.

It is important to highlight that the descriptive statistics for all assets were computed based on a dataset comprising 379 days of data. However, the asset ONDO presents a slight deviation, as it has only been available for 330 days (the asset was launched 330 days ago). This discrepancy is worth noting, as it may affect the reliability of the statistical analysis for ONDO compared to the other assets included in the study.

However, since the difference between 330 and 379 days is relatively small, we believe this deviation can be neglected for the purposes of this study. The difference of just 49 days is unlikely to significantly distort the overall trends or findings.

# Table 2. Summary of descriptive statistics for selected cryptocurrency assets.

| Alt | Name | Coin/Token | Count | Mean | SD | Min | 25% | 50% | 75% | Max | Variance |
|---|---|---|---|---|---|---|---|---|---|---|---|
| 1 | ETH | Ethereum | 379 | 0.0023 | 0.0334 | -0.1006 | -0.0148 | 0.0022 | 0.0198 | 0.1905 | 0.0011 |
| 2 | SOL | Solana | 379 | 0.0045 | 0.0446 | -0.1362 | -0.0253 | 0.0001 | 0.0337 | 0.1442 | 0.0020 |
| 3 | TRX | Tron | 379 | 0.0037 | 0.0520 | -0.2090 | -0.0081 | 0.0016 | 0.0120 | 0.8888 | 0.0027 |
| 4 | BNB | BSC (Binance) | 379 | 0.0035 | 0.0304 | -0.0852 | -0.0131 | 0.0019 | 0.0173 | 0.1651 | 0.0009 |
| 5 | BTC | Bitcoin | 379 | 0.0030 | 0.0275 | -0.0824 | -0.0110 | 0.0017 | 0.0158 | 0.1227 | 0.0008 |
| 6 | LINK | Chainlink | 379 | 0.0029 | 0.0467 | -0.1504 | -0.0264 | -0.0004 | 0.0280 | 0.3219 | 0.0022 |
| 7 | ARB | Arbitrum | 379 | 0.0013 | 0.0502 | -0.1712 | -0.0272 | -0.0022 | 0.0263 | 0.2444 | 0.0025 |
| 8 | SUI | Sui | 379 | 0.0073 | 0.0624 | -0.1656 | -0.0319 | 0.0002 | 0.0314 | 0.3931 | 0.0039 |
| 9 | AVAX | Avalanche | 379 | 0.0073 | 0.0624 | -0.1656 | -0.0319 | 0.0002 | 0.0314 | 0.3931 | 0.0039 |
| 10 | POL | Polygon | 379 | 0.0004 | 0.0446 | -0.1703 | -0.0268 | -0.0022 | 0.0245 | 0.1536 | 0.0020 |
| 11 | CRV | Curve | 379 | 0.0035 | 0.0579 | -0.2035 | -0.0276 | -0.0010 | 0.0311 | 0.2730 | 0.0033 |
| 12 | APT | Aptos | 379 | 0.0032 | 0.0522 | -0.1761 | -0.0273 | -0.0025 | 0.0279 | 0.2484 | 0.0027 |
| 13 | OP | Optimism | 379 | 0.0028 | 0.0590 | -0.1647 | -0.0344 | -0.0015 | 0.0354 | 0.3937 | 0.0035 |
| 14 | CORE | CORE | 379 | 0.0067 | 0.0959 | -0.2911 | -0.0344 | -0.0035 | 0.0259 | 0.8688 | 0.0092 |
| 15 | UNI | Uniswap | 379 | 0.0045 | 0.0597 | -0.1410 | -0.0259 | 0.0004 | 0.0270 | 0.5379 | 0.0036 |
| 16 | MNT | Mantle | 379 | 0.0032 | 0.0458 | -0.1012 | -0.0223 | -0.0007 | 0.0208 | 0.3299 | 0.0021 |
| 17 | CRO | Cronos | 379 | 0.0032 | 0.0532 | -0.1483 | -0.0202 | -0.0019 | 0.0201 | 0.6701 | 0.0028 |
| 18 | ONDO | Ondo | 330 | 0.0088 | 0.0046 | 0.0677 | -0.0329 | 0.0003 | 0.0438 | 0.3836 | 0.1439 |
| 19 | NUM | Numbers | 379 | 0.0074 | 0.1118 | -0.2281 | -0.0408 | -0.0093 | 0.0312 | 1.1497 | 0.0125 |
| 20 | CAKE | Pancakeswap | 379 | 0.0022 | 0.0469 | -0.1965 | -0.0228 | 0.0006 | 0.0234 | 0.2104 | 0.0022 |
| 21 | MKR | Maker | 379 | 0.0019 | 0.0459 | -0.1455 | -0.0235 | -0.0021 | 0.0205 | 0.2367 | 0.0021 |
| 22 | RUNE | Thorchain | 379 | 0.0016 | 0.0572 | -0.1824 | -0.0352 | -0.0017 | 0.0350 | 0.3148 | 0.0033 |
| 23 | TON | TON | 379 | 0.0035 | 0.0443 | -0.1497 | -0.0207 | 0.0002 | 0.0223 | 0.2294 | 0.0020 |
| 24 | ADA | Cardano | 379 | 0.0039 | 0.0449 | -0.1588 | -0.0212 | 0.0006 | 0.0248 | 0.2276 | 0.0020 |
| 25 | GNO | Gnosis | 379 | 0.0020 | 0.0418 | -0.1232 | -0.0224 | 0.0006 | 0.0241 | 0.2099 | 0.0017 |
| 26 | AAVE | Aave | 379 | 0.0048 | 0.0501 | -0.1725 | -0.0254 | 0.0002 | 0.0282 | 0.2823 | 0.0025 |
| 27 | AR | Arweave | 379 | 0.0051 | 0.0696 | -0.2026 | -0.0399 | -0.0019 | 0.0403 | 0.5116 | 0.0048 |
| 28 | DYDX | dYdX | 379 | 0.0004 | 0.0541 | -0.2204 | -0.0305 | -0.0017 | 0.0266 | 0.3138 | 0.0029 |
| 29 | NEAR | Near | 379 | 0.0052 | 0.0605 | -0.1712 | -0.0315 | -0.0014 | 0.0379 | 0.3681 | 0.0037 |
| 30 | 1INCH | 1inch | 379 | 0.0023 | 0.0502 | -0.2352 | -0.0281 | 0.0024 | 0.0295 | 0.2447 | 0.0025 |
| 31 | ROSE | Oasis | 379 | 0.0020 | 0.0513 | -0.1951 | -0.0316 | -0.0040 | 0.0337 | 0.1863 | 0.0026 |
| 32 | SEI | Sei | 379 | 0.0046 | 0.0682 | -0.1820 | -0.0373 | -0.0057 | 0.0375 | 0.2713 | 0.0047 |
| 33 | ONE | Harmony | 379 | 0.0041 | 0.0560 | -0.1984 | -0.0299 | 0.0020 | 0.0369 | 0.2582 | 0.0031 |
| 34 | MANA | Decentraland | 379 | 0.0023 | 0.0503 | -0.1877 | -0.0226 | 0.0021 | 0.0254 | 0.3839 | 0.0025 |
| 35 | KAVA | Kava | 379 | 0.0003 | 0.0436 | -0.2042 | -0.0222 | 0.0022 | 0.0256 | 0.1779 | 0.0019 |
| 36 | RON | Ronin | 379 | 0.0031 | 0.0504 | -0.1811 | -0.0287 | 0.0007 | 0.0318 | 0.1700 | 0.0025 |
| 37 | VET | Vechain | 379 | 0.0040 | 0.0516 | -0.1661 | -0.0256 | -0.0001 | 0.0302 | 0.3717 | 0.0027 |
| 38 | LTC | Litecoin | 379 | 0.0022 | 0.0381 | -0.1838 | -0.0163 | 0.0020 | 0.0197 | 0.1863 | 0.0014 |
| 39 | EOS | EOS | 379 | 0.0023 | 0.0461 | -0.2158 | -0.0218 | 0.0024 | 0.0226 | 0.2390 | 0.0021 |
| 40 | CELO | Celo | 379 | 0.0028 | 0.0566 | -0.1891 | -0.0292 | 0.0004 | 0.0294 | 0.5029 | 0.0032 |
| 41 | FTM | Fantom | 379 | 0.0057 | 0.0605 | -0.1893 | -0.0365 | 0.0041 | 0.0433 | 0.2945 | 0.0037 |
| 42 | EGLD | MultiversX | 379 | 0.0012 | 0.0439 | -0.1943 | -0.0243 | -0.0014 | 0.0276 | 0.1340 | 0.0019 |
| 43 | STX | Stacks | 379 | 0.0050 | 0.0618 | -0.1726 | -0.0329 | -0.0007 | 0.0358 | 0.4108 | 0.0038 |
| 44 | XMR | Monero | 379 | 0.0013 | 0.0382 | -0.3663 | -0.0126 | 0.0024 | 0.0168 | 0.2379 | 0.0015 |
| 45 | ATOM | Cosmos | 379 | 0.0009 | 0.0426 | -0.1692 | -0.0229 | -0.0017 | 0.0221 | 0.1887 | 0.0018 |
| 46 | XTZ | Tezos | 379 | 0.0027 | 0.0504 | -0.1824 | -0.0204 | 0.0001 | 0.0230 | 0.4876 | 0.0025 |
| 47 | ALGO | Algorand | 379 | 0.0044 | 0.0525 | -0.1587 | -0.0260 | 0.0019 | 0.0288 | 0.3759 | 0.0028 |

Source: Authors' own compilation based on data from CoinGecko.com

Therefore, we think it is beneficial to consider ONDO as an option rather than removing it entirely, as its inclusion may offer valuable insights, particularly as the asset develops further and additional data becomes available. Removing it could limit the scope of our analysis, especially considering that many portfolio optimization studies may encounter similar data limitations with newly launched assets.

**3.1.6. Criteria for evaluating cryptocurrency performance.** This part is dedicated to the description of the criteria for evaluating cryptocurrency performance. In this paper, we introduce 11 comprehensive criteria for assessing crypto assets, as detailed in Table 3. This table not only presents the criteria themselves but also outlines the types of benefits and costs associated with each criterion, thereby providing a clear and structured framework for evaluation. To the best of our knowledge, no existing study has implemented a systematic pre-selection process specifically in the context of cryptocurrency portfolio optimization. This gap has resulted in a lack of established benchmarks for these criteria, making it challenging for researchers and practitioners to assess cryptocurrency assets effectively. To address this issue, we aim to establish a robust set of criteria that can serve as a valuable benchmark for evaluating cryptocurrencies during the pre-selection phase. Our criteria are designed to ensure that the assessment of cryptocurrency performance is both rigorous and comprehensive, addressing the unique challenges posed by this dynamic asset class. By developing these criteria, we hope to facilitate more informed decision-making in the rapidly evolving cryptocurrency market. This systematic approach not only enhances the reliability of our evaluations but also contributes to the broader field of cryptocurrency research by providing a structured methodology that others can replicate.

**Table 3. Comprehensive criteria for evaluating cryptocurrency performance.**

| Criteria | Cost/ Benefit | Description |
|---|---|---|
| Avg TPS | Benefit | The average number of transactions processed per second (TPS) on a blockchain network. This metric indicates the network's speed and capacity to handle transactions. |
| Avg Active Addresses | Benefit | The average number of addresses that participate in network transactions within a specific time frame (in this case, daily). It counts addresses that have conducted at least one transaction on the network. |
| Max Drawdown From ATH | Cost | The maximum percentage decline in the value of an asset from its all-time high (ATH). This metric is considered a measure of cost and risk, demonstrating how much a project's price can drop during bearish market conditions. |
| Avg TVL (Billion Dollars) | Benefit | The average total value locked (TVL) in a decentralized finance (DeFi) protocol. It indicates how much value (typically cryptocurrencies) is locked in smart contracts on a platform. |
| Avg Fee (Dollars) | Cost | The average transaction fee on a blockchain network, referring to the cost's users incur to execute transactions. These fees are typically paid to miners or validators as rewards for processing and validating transactions. Transaction fees can vary depending on network congestion, transaction size, and user prioritization. |
| Avg Daily Volume (Billion Dollars) | Benefit | The average daily trading volume of a cryptocurrency. This metric includes buy and sell transactions across all trading platforms (both centralized and decentralized exchanges). |
| Avg M-Cap (Billion Dollars) | Benefit | The average total market value of a project in the cryptocurrency market. Market cap is calculated by multiplying the total circulating supply of tokens by the current price per token. |
| Total Revenue (Million Dollars) | Benefit | Refers to the total income generated by a blockchain or protocol through various activities, such as transaction fees, service fees, and other financial operations. This revenue is typically collected from users or participants within the ecosystem. |
| Circulating/Total Supply | Benefit | This ratio represents the proportion of tokens in circulation relative to the total token supply. It indicates what percentage of a cryptocurrency's total supply is currently available and tradable in the market. |
| Full-Time Developers | Benefit | Refers to developers who work full-time on the development, improvement, and maintenance of a blockchain project. These individuals are part of the project's technical team and consistently write, update, and test the project's code. This metric reflects the strength of the development team. |
| Total Developers | Benefit | Refers to the total number of developers involved in the development of a blockchain project, whether full-time, part-time, or volunteer. This metric includes all contributors who have updated or participated in the project's code repositories, such as those on GitHub. |

Note: "Ave"=Average, "Max"=Maximum

Source: Authors' own compilation

Next, we employed the Analytic Hierarchy Process (AHP) method [73] to determine the weight of each criterion involved in the evaluation of cryptocurrency performance. AHP is a structured decision-making tool that allows for the systematic comparison of multiple criteria by breaking down complex problems into a hierarchy of manageable components. This method facilitates a thorough assessment by enabling decision-makers to evaluate the relative importance of each criterion through pairwise comparisons. The structure of the AHP model is illustrated in Fig 2, which visually represents the hierarchical arrangement of the criteria. This figure outlines how the criteria are organized, showing the relationships and dependencies among them. At the top of the hierarchy is the overall goal, followed by the criteria that contribute to this goal.

The results of the AHP analysis are presented in Table 4, highlighting the weights assigned to each criterion based on pairwise comparisons.

In total, 55 comparisons were made during the AHP process, resulting in a Consistency Ratio (CR) of 1.5%, indicating a high level of consistency in the judgments made. The principal eigenvalue calculated from the AHP model was found to be 11.221, further confirming the reliability of the derived weights. This systematic approach ensures that the evaluation of cryptocurrency assets is both rigorous and justified, enhancing the robustness of our framework.

### 3.2. Stage 2 - Portfolio optimization

After introducing the pre-selection process (Stage 1) in the previous section, this section focuses on the optimization process. Given that uncertainty is a fundamental characteristic of capital markets, this paper integrates credibility theory with the CVaR framework, effectively leveraging their combined strengths to model downside risk and manage uncertainty during the optimization process of Stage 2. Furthermore, practical constraints commonly encountered in real-world investment scenarios are taken into account, ensuring that the proposed framework is both theoretically robust and practically applicable within the dynamic cryptocurrency market. This integrated model is employed for constructing portfolios, facilitating more informed investment decisions.

In this section, we will first present key concepts and definitions related to fuzzy theory and credibility measures, providing a foundational understanding necessary for the subsequent discussion. Following this, we will describe the proposed model in detail, elucidating how it addresses the challenges of uncertainty and risk management in cryptocurrency portfolio optimization.

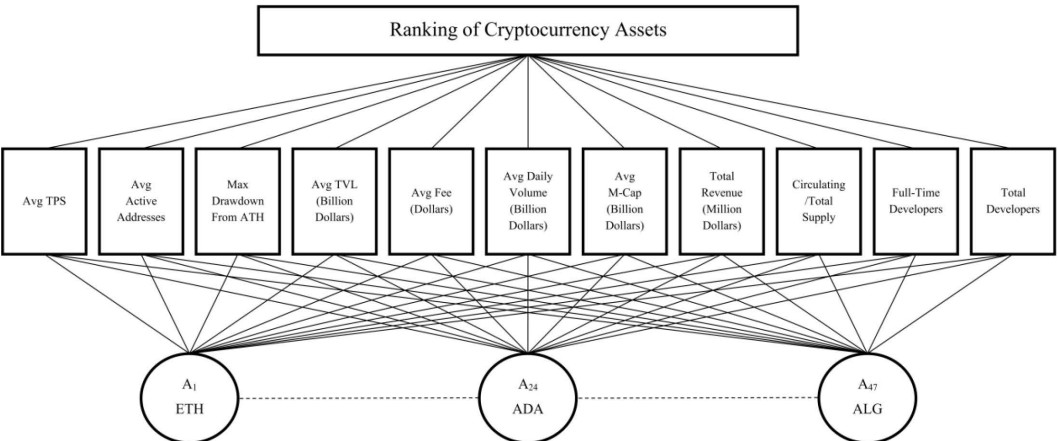

**Fig 2. Schematic representation of the AHP model for weighting evaluating cryptocurrency criteria.**

**Table 4. Weights assigned to each criterion based on AHP method.**

| Code | Criteria | Weight |
|---|---|---|
| $C_1$ | Avg TPS | 11.10% |
| $C_2$ | Avg Active Addresses | 9.30% |
| $C_3$ | Max Drawdown From ATH | 3.30% |
| $C_4$ | Avg TVL (Billion Dollars) | 17.80% |
| $C_5$ | Avg Fee (Dollars) | 10.90% |
| $C_6$ | Avg Daily Volume (Billion Dollars) | 6.40% |
| $C_7$ | Avg M-Cap (Billion Dollars) | 9.00% |
| $C_8$ | Total Revenue (Million Dollars) | 18.10% |
| $C_9$ | Circulating/Total Supply | 4.30% |
| $C_{10}$ | Full-Time Developers | 2.50% |
| $C_{11}$ | Total Developers | 7.20% |

Source: Authors' own computation from AHP method

**3.2.1. *Some concepts and definitions in fuzzy theory and credibility measure*.** In this section, we will revisit several fundamental concepts and definitions that are crucial for understanding the subsequent content. We will specifically focus on three key areas: (I) Fuzzy set theory, (II) Fuzzy numbers, and (III) Credibility theory. Together, these elements form an integral part of the analytical framework that supports this study.

Definition 1. Fuzzy set theory

Classical set theory evaluates elements based on binary criteria, determining whether an element is either a member of a set or not. This black-and-white approach can be limiting, as it does not account for the complexities of real-world situations where membership may not be absolute. In contrast, fuzzy set theory offers a more sophisticated mathematical framework that allows for a nuanced evaluation of element membership within a set. In fuzzy set theory, this evaluation is represented through a membership function, which assigns each element a membership grade that ranges from zero to one. This functionality enables the model to effectively capture the vagueness and uncertainty inherent in various scenarios, allowing for a more flexible representation of reality.

Let $X$ denote a universe of discourse, with its generic element represented as $x$. A fuzzy set $A$ can be characterized as a collection of ordered pairs defined within the universe $X$, expressed mathematically as follows:

$$A = \left\{ (x, \mu_A(x)| \ x \in X) \right\} \tag{1}$$

In this expression, $\mu_A(x)$ denotes the membership function, which quantifies the degree of membership of an element $x \in X$. The membership function is defined within the real interval [0,1]. A value of $\mu_A(x) = 1$ indicates full membership, meaning that the element completely belongs to the fuzzy set A, while a value of $\mu_A(x) = 0$ signifies no membership. Intermediate values represent varying degrees of membership, reflecting how strongly the element $x$ belongs to the fuzzy set A. This flexibility to express partial membership is particularly useful in domains where uncertainty and ambiguity are prevalent, such as in decision-making processes, risk assessment, and various applications in economics, finance and engineering. By accommodating the complexities of real-world situations, fuzzy set theory enhances our ability to model and analyze data in a way that traditional set theory cannot.

Definition 2. Fuzzy Numbers

In practical applications, it is common for experts to express their assessments and judgments using fuzzy numbers. These fuzzy numbers are particularly useful in situations where uncertainty or imprecision is inherent in the data or the information being conveyed. Fuzzy numbers allow for a more flexible representation of values, enabling decision-makers to capture the nuances of their evaluations. Two specific forms of fuzzy numbers that are widely used are triangular and trapezoidal fuzzy numbers. These forms are defined by their unique characteristics as follows:

*Triangular fuzzy number* Let $\widetilde{A}$ be defined as a triangular fuzzy number characterized by the parameters $(a_1,\ a_2, a_3)$, where $a_1$, $a_2$, and $a_3$ are real numbers satisfying $a_1 \leq a_2 \leq a_3$. The membership function $\mu_A(x)$ associated with $\widetilde{A}$ is defined piecewise as follows:

$$\mu_{\widetilde{A}}(x) = \begin{cases} 0, & x \in (-\infty, a_1) \\ \frac{x-a_1}{a_2-a_1}, & x \in [a_1, a_2] \\ \frac{a_3-x}{a_3-a_2}, & x \in [a_2, a_3] \\ 0, & x \in (a_3, +\infty) \end{cases}$$

(2)

In this representation, the triangular fuzzy number $\widetilde{A}$ achieves its maximum membership value of 1 at $x = a_2$. As $x$ moves away from $a_2$ towards either $a_1$ or $a_3$, the membership values decrease linearly to 0, creating the characteristic triangular shape depicted in Fig 3.

*Trapezoid fuzzy number* Let $\widetilde{A}$ be defined as a trapezoid fuzzy number characterized by the parameters $(a_1,\ a_2, a_3, a_3)$, where $a_1$, $a_2$, $a_3$ and $a_4$ are real numbers satisfying $a_1 \leq a_2 \leq a_3 \leq a_4$. The membership function $\mu_A(x)$ associated with $\widetilde{A}$ is defined piecewise as follows:

$$\mu_{\widetilde{A}}(x) = \begin{cases} 0, & x \in (-\infty, a_1) \\ \frac{x-a_1}{a_2-a_1}, & x \in [a_1, a_2] \\ 1, & x \in [a_2, a_3] \\ \frac{a_4-x}{a_4-a_3}, & x \in [a_3, a_4] \\ 0, & x \in (a_4, +\infty) \end{cases}$$

(3)

The membership function for a trapezoidal fuzzy number takes on a trapezoidal shape, where the fuzzy number $\widetilde{A}$ maintains a constant maximum membership value of 1 over the interval $[a_2, a_3]$. Outside this interval, the membership values decrease linearly to 0 as $x$ approaches $a_1$ or $a_4$, resulting in the characteristic trapezoidal shape depicted in Fig 4.

Definition 3. Credibility theory

Credibility theory, originally introduced by Liu [23] and further developed in subsequent works [24], provides a robust mathematical framework for analyzing and modeling fuzzy phenomena. This theory is essential for the advancement of

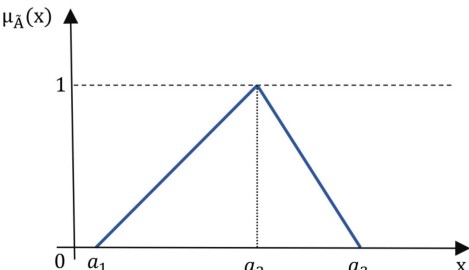

**Fig 3. Visual representation of a triangular fuzzy number.**

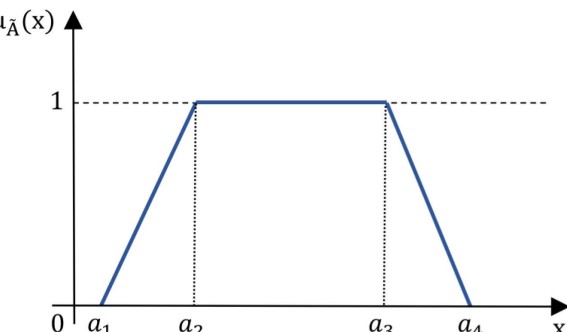

**Fig 4. Visual representation of a trapezoid fuzzy number.**

credibility fuzzy programming, which allows decision-makers to quantify and evaluate their level of confidence in meeting specific constraints amid uncertainty. Credibility theory offers a versatile approach to managing various types of fuzzy data, such as triangular and trapezoidal fuzzy numbers, which are commonly employed to represent uncertain information in decision-making processes. By incorporating these fuzzy numbers, the theory facilitates a more nuanced and realistic portrayal of uncertainty encountered in real-world situations. Liu [23,24] lays out the fundamental definitions and notations that form the basis of credibility theory, including the concept of the credibility measure. This measure is a crucial tool for assessing the likelihood or degree of confidence that a particular event or condition will occur within a fuzzy context. As highlighted by Liu and Liu [74], the calculation of the credibility measure is expressed as follows:

$$Cr\{\xi \in A\} = \frac{1}{2}(Pos\{\xi \in A\} + Nes\{\xi \in A\})$$

(4)

In this equation, $Pos\{\xi \in A\}$ denotes the possibility measure of the event $\{\xi \in A\}$, while $Nes\{\xi \in A\}$ represents the necessity measure of the same event. Both measures are fundamental concepts in fuzzy set theory:

- *Possibility Measure* (*Pos*): The possibility measure, denoted as $Pos\{\xi \in A\}$, evaluates the highest degree of membership of the variable $\xi$ within the fuzzy set $A$. This measure reflects the most plausible extent to which the event $\{\xi \in A\}$ can occur, capturing the potential for membership in a fuzzy context. It is mathematically defined as:

$$Pos\{\xi \in A\} = sup\ \mu(x)_{x \in A}$$

(5)

Here, $sup\ \mu(x)_{x \in A}$ signifies the supremum (or least upper bound) of the membership function $\mu(x)$ for all $x$ within the set $A$. This measure essentially provides a value between 0 and 1, indicating the highest degree to which $\xi$ can be considered a member of $A$.

- *Necessity Measure* (*Nes*): In contrast, the necessity measure $Nes\{\xi \in A\}$ assesses the degree of certainty associated with the event $\{\xi \in A\}$. It is calculated as the complement of the highest degree of membership found in the complement set $A^c$, providing a measure of how definitive the membership is. This is expressed as:

$$Nes\{\xi \in A\} = 1 - \sup \mu(x)_{x \in A^c}$$

(6)

In this formulation, $\sup \mu(x)_{x \in A^c}$ represents the maximum degree of membership for any element outside the set $A$. By taking the complement, this measure effectively captures the certainty that $\xi$ belongs to $A$.

Since $Pos\{\xi \in A\} = sup\ \mu(x)_{x \in A}$ and $Nes\{\xi \in A\} = 1 - \sup \mu(x)_{x \in A^c}$, the credibility measure can also be formulated as:

$$Cr\{\xi \in A\} = \frac{1}{2}(sup\ \mu(x)_{x \in A} + 1 - \sup \mu(x)_{x \in A^c})$$

(7)

This formulation illustrates that the credibility measure takes into account both the possibility and necessity of the event $\{\xi \in A\}$, offering a thorough evaluation of its likelihood within a fuzzy context.

When examining a specific fuzzy event characterized by $\{\xi \leq r\}$, where $r$ is a real number, the credibility measure is expressed as:

$$Cr\{\xi \leq r\} = \frac{1}{2}(sup\ \mu(x)_{x \leq r} + 1 - \sup \mu(x)_{x > r})$$

(8)

This equation allows for the assessment of the credibility that a fuzzy variable $\xi$ will assume a value less than or equal to a specified real number $r$. It considers both the maximum membership value within the interval $(-\infty, r]$ and the complement membership value in the interval $(r, +\infty)$, providing a balanced perspective on the likelihood of the event.

Another key concept in credibility theory is the expected value of a fuzzy variable $\xi$, which represents the "average" outcome of $\xi$ while accounting for its fuzzy characteristics. It is calculated using the following expression:

$$E[\xi] = \int_0^{+\infty} Cr\{\xi \geq r\}\,dr - \int_{-\infty}^0 Cr\{\xi \leq r\}\,dr$$

(9)

Equation (9) integrates the credibility measures across the entire real line, effectively determining the expected value of $\xi$ by weighing contributions from both positive and negative ranges. The first integral evaluates the credibility of $\xi$ being greater than or equal to $r$, while the second integral assesses the credibility of $\xi$ being less than or equal to $r$.

The following sections of the paper will concentrate on applying the credibility measure to specific types of fuzzy variables, specifically triangular and trapezoidal fuzzy numbers.

*Credibility measure for triangular fuzzy numbers* Consider a fuzzy variable characterized by the triplet $(a_1, a_2, a_3)$ of crisp numbers, where $(a_1 \leq a_2 \leq a_3)$. Utilizing the general formula for credibilistic expected value (as presented in Equation (9)), we can derive the credibilistic expected value of the triangular fuzzy variable $\xi$ as follows:

$$E[\xi] = \frac{a_1 + 2a_2 + a_3}{4}$$

(10)

This expression indicates that the expected value of the triangular fuzzy variable is a weighted average, giving more significance to the mode $a_2$ due to its central position within the triangular structure.

Next, we determine the credibility measure $Cr\{\xi \leq r\}$ for a triangular fuzzy number using the following equation:

$$Cr\{\xi \leq r\} = \begin{cases} 0, & r \leq a_1 \\ \frac{r-a_1}{2(a_2-a_1)}, & a_1 \leq r \leq a_2 \\ \frac{a_3-2a_2+r}{2(a_3-a_2)}, & a_2 \leq r \leq a_3 \\ 1, & a_3 \leq r \end{cases}$$

(11)

This measure captures the probability that the fuzzy variable $\xi$ will take on a value less than or equal to a specific real number $r$. It incorporates different cases depending on the value of $r$ relative to the parameters $a_1, a_2,$ and $a_3$.

Similarly, the credibility measure $Cr\{\xi \geq r\}$ for a triangular fuzzy number can be expressed as:

$$Cr\{\xi \geq r\} = \begin{cases} 1, & r \leq a_1 \\ \frac{2a_2 - a_1 - r}{2(a_2 - a_1)}, & a_1 \leq r \leq a_2 \\ \frac{a_3 - r}{2(a_3 - a_2)}, & a_2 \leq r \leq a_3 \\ 0, & a_3 \leq r \end{cases}$$

(12)

This measure assesses the probability that the fuzzy variable $\xi$ will assume a value greater than or equal to $r$. Like the previous measure, it accounts for different scenarios based on the position of $r$ relative to the triangular parameters.

Fig 5 visually represents these credibility measures, illustrating the likelihood of various outcomes in the context of a triangular fuzzy variable. The graphical depiction aids in understanding how the credibility measures change with different values of $r$ and highlights the inherent uncertainty associated with fuzzy variables.

*Credibility measure for trapezoidal fuzzy numbers* Consider a fuzzy variable defined by the quadruplet $(a_1, a_2, a_3, a_4)$ of crisp numbers, where $(a_1 \leq a_2 \leq a_3 \leq a_4)$. Using the general formula for the credibilistic expected value (as outlined in Equation (9)), we can define the expected value for a trapezoidal fuzzy variable $\xi$ as follows:

$$E[\xi] = \frac{a_1 + a_2 + a_3 + a_4}{4}$$

(13)

This equation indicates that the expected value of the trapezoidal fuzzy variable is the arithmetic mean of its four parameters, reflecting the central tendency of the fuzzy number.

Next, we determine the credibility measure $Cr\{\xi \leq r\}$ for a trapezoidal fuzzy number, which is expressed as:

$$Cr\{\xi \leq r\} = \begin{cases} 0, & r \leq a_1 \\ \frac{r - a_1}{2(a_2 - a_1)}, & a_1 \leq r \leq a_2 \\ \frac{1}{2}, & a_2 \leq r \leq a_3 \\ \frac{a_4 - 2a_3 + r}{2(a_4 - a_3)}, & a_3 \leq r \leq a_4 \\ 1, & a_4 \leq r \end{cases}$$

(14)

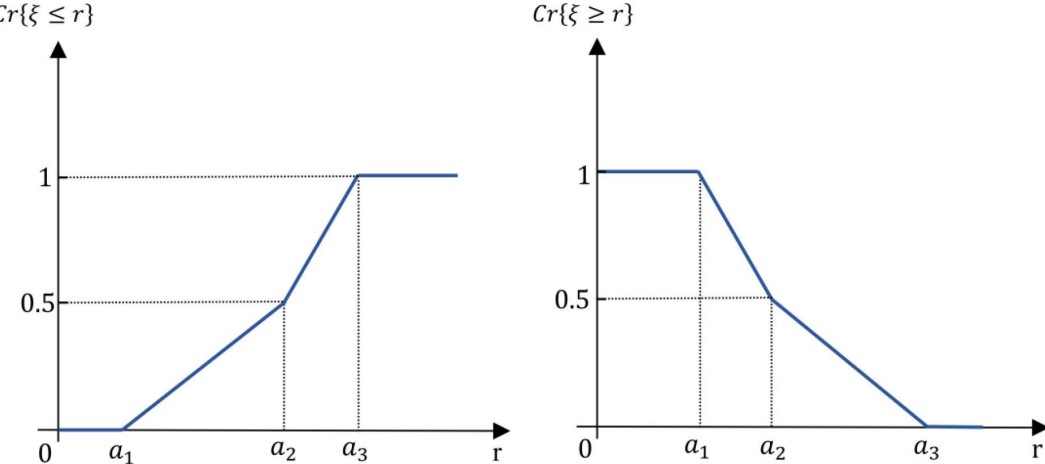

**Fig 5. Credibility measures for a triangular fuzzy variable.**

This measure evaluates the probability that the fuzzy variable $\xi$ will take on a value less than or equal to a specific real number $r$. It considers various cases based on the position of $r$ relative to the trapezoidal parameters, reflecting the inherent uncertainty associated with trapezoidal fuzzy numbers.

In a similar fashion, the credibility measure $Cr\{\xi \geq r\}$ for a trapezoidal fuzzy number can be formulated as follows:

$$Cr\{\xi \geq r\} = \begin{cases} 1, & r \leq a_1 \\ \frac{2a_2 - r - a_1}{2(a_2 - a_1)}, & a_1 \leq r \leq a_2 \\ \frac{1}{2}, & a_2 \leq r \leq a_3 \\ \frac{a_4 - r}{2(a_4 - a_3)}, & a_3 \leq r \leq a_4 \\ 0, & a_4 \leq r \end{cases}$$

(15)

This measure assesses the probability that the fuzzy variable $\xi$ will assume a value greater than or equal to $r$. Like the previous measure, it accounts for different scenarios depending on the value of $r$ in relation to the trapezoidal parameters.

Fig 6 visually illustrates these credibility measures, depicting the likelihood of various outcomes in the context of a trapezoidal fuzzy variable. The graphical representation aids in understanding how the credibility measures vary with different values of $r$, highlighting the uncertainties and potentialities associated with trapezoidal fuzzy numbers.

**3.2.2. *Proposed model*.** In this paper, we employ the Credibilistic CVaR criterion, taking into account practical constraints such as cardinality constraints, and floor and ceiling constraints – also known as quantity, threshold, or box constraints. We utilize trapezoidal fuzzy variables to model this Credibilistic CVaR.

In this section, we will detail the model and the modeling process. We begin by introducing the concept of CVaR and explaining its significance in risk management. Following that, we will delve into the specifics of Credibilistic CVaR, highlighting its advantages and how it accommodates uncertainty. Next, we will discuss the various constraints involved in our model, including cardinality constraints that limit the number of selected elements, as well as floor and ceiling constraints that set limits on variable quantities. Finally, we will present the complete model, integrating the Credibilistic CVaR with the identified constraints, to provide a comprehensive framework for decision-making under uncertainty.

**3.2.3. Conditional Value at Risk (CVaR).** A key objective of risk management is to evaluate and enhance the performance of financial investments by thoroughly analyzing the associated risks involved in profit generation. One of the most commonly employed methods for quantifying these risks is VaR, which has gained widespread acceptance as a standard tool for estimating potential losses. However, despite its widespread use, VaR has significant limitations,

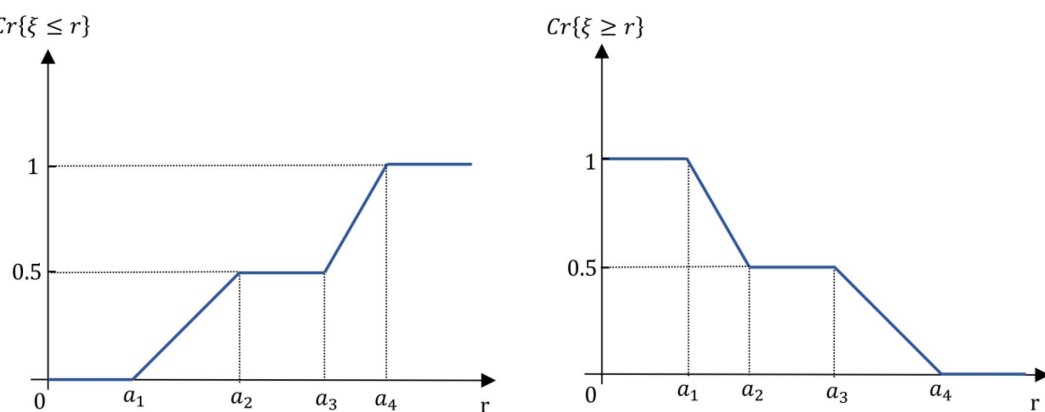

**Fig 6. Credibility measures for a trapezoidal fuzzy variable.**

particularly regarding its inability to adequately capture extreme or tail risks—those rare but severe losses that can occur in adverse market conditions. In light of these shortcomings, CVaR has emerged as a more advanced and comprehensive risk assessment measure. Often referred to as mean excess loss, mean shortfall, or tail VaR, CVaR offers a more nuanced understanding of risk by concentrating on the potential losses that surpass the VaR threshold. This focus allows CVaR to provide valuable insights into the tail end of the loss distribution, making it especially relevant in situations where extreme risks are a primary concern.

CVaR is a significant risk assessment tool that goes beyond traditional VaR by concentrating on the tail end of the loss distribution. While VaR provides a measure of the maximum expected loss within a specified confidence level, it falls short by not fully encompassing the potential extreme losses that can occur beyond this threshold. In contrast, CVaR captures the expected loss that exceeds the VaR level, thus offering a more comprehensive perspective on the risks associated with extreme outcomes. This characteristic makes CVaR particularly valuable in situations where tail risk is a critical concern, such as in financial risk management, insurance underwriting, and portfolio optimization strategies. By focusing on the worst-case scenarios, CVaR enables decision-makers to better understand and mitigate the risks of substantial losses. One of the fundamental advantages of CVaR is its coherence, which means it adheres to several desirable mathematical properties, including sub-additivity, translation invariance, monotonicity, and positive homogeneity. The property of sub-additivity is especially important, as it ensures that the risk of a combined portfolio does not exceed the sum of the individual portfolio risks, thereby encouraging diversification. These properties collectively enhance the robustness and reliability of CVaR as a risk assessment measure across various financial contexts. Mathematically, the CVaR of a random variable ξ at a confidence level $\alpha$ can be expressed as:

$$CVaR(x, \eta) = \eta + (1-\alpha)^{-1} \int_{\xi \varepsilon R^n} [f(X, \xi) - \eta]^+ p(\xi) d\xi$$

(16)

In this formulation, $[f(X, \xi) - \eta]^+$ is expressed as:

$$[f(X, \xi) - \eta]^+ \stackrel{\text{def}}{=\joinrel=} \begin{cases} f(X, \xi) - \eta & \text{if} \quad f(X, \xi) - \eta > 0 \\ 0 & \text{if} \quad f(X, \xi) - \eta \leq 0 \end{cases}$$

(17)

In this context, $\eta$ represents the VaR threshold, $\alpha$ denotes the confidence level, $f(X, \xi)$ indicates the loss function, and $p(\xi)$ is the probability density function for $\xi$. This formulation utilizes linear programming techniques to efficiently compute CVaR, making it highly applicable in financial optimization scenarios where understanding and managing risk is critical.

**3.2.4. Credibilistic VaR and CVaR.** Integrating credibility theory into the computation of CVaR enables a more flexible and nuanced approach to managing uncertainty. Credibility theory is particularly valuable in contexts where data is fuzzy or imprecise, offering an alternative to traditional probability-based methods. By applying this framework, practitioners can better account for the inherent uncertainties in their risk assessments.

Since credibilistic CVaR can be derived from credibilistic VaR, we first need to introduce the concept of credibilistic VaR to facilitate this process. For a fuzzy variable $\xi$ and a confidence level $\alpha \in (0, 1]$, VaR within the context of credibility theory can be defined as:

$$\xi_{VaR}(\alpha) = -sup\{x \mid Cr\{\xi \leq x\} \leq \alpha\}$$

(18)

This definition helps identify the maximum value of $x$ for which the credibility measure $Cr\{\xi \leq x\}$ is less than or equal to the specified confidence level $\alpha$. In essence, it provides a fuzzy counterpart to the traditional VaR measure, adapting the concept to better handle uncertainty and imprecision inherent in fuzzy data. Moreover, there is an alternative formulation for VaR within the framework of credibility theory:

$$\xi_{VaR}(\alpha) = -inf\left\{x|Cr\left\{\xi \leq x\right\} \geq \alpha\right\} = -inf\left\{x|\Phi(x) \geq \alpha\right\} = -\Phi^{-1}(\alpha) \tag{19}$$

In this expression, $\Phi(x)$ denotes the cumulative credibility distribution function.

Based on the information provided, the VaR for a triangular fuzzy variable defined by the parameters $\xi = (a_1,\ a_2,\ a_3)$ and a confidence level $\alpha \in (0,\ 1]$ can be determined using the following equations:

$$\xi_{VaR}(\alpha) = \begin{cases} 2\left(a_1 - a_2\right)\alpha - a_1 & \alpha \in (0,\ 0.5] \\ 2\left(a_2 - a_3\right)\alpha + a_3 - 2a_2 & \alpha \in (0.5,\ 1] \end{cases} \tag{20}$$

Similarly, the VaR for a trapezoidal fuzzy variable defined by the parameters $\xi = (a_1,\ a_2,\ a_3, a_4)$ and a confidence level $\alpha \in (0,\ 1]$ can be determined using the following equations:

$$\xi_{VaR}(\alpha) = \begin{cases} 2\left(a_1 - a_2\right)\alpha - a_1 & \alpha \in (0,\ 0.5] \\ 2\left(a_3 - a_4\right)\alpha + a_4 - 2a_3 & \alpha \in (0.5,\ 1] \end{cases} \tag{21}$$

The CVaR within the framework of credibility theory, referred to as $\xi_{CVaR}$, is derived by integrating the VaR function across the confidence interval:

$$\xi_{CVaR} = \frac{1}{1 - \alpha} \int_{\alpha}^{1} \xi_{VaR}(r)dr \tag{22}$$

Based on the information provided, the CVaR for a triangular fuzzy variable defined by the parameters $\xi = (a_1,\ a_2,\ a_3)$ and a confidence level $\alpha \in (0,\ 1]$ can be determined using the following equations:

$$\xi_{CVaR}(\alpha) = \begin{cases} \alpha a_1 - (1 + \alpha)a_2 & \alpha \in (0,\ 0.5] \\ (\alpha - 1)a_2 - \alpha a_3 & \alpha \in (0.5,\ 1] \end{cases} \tag{23}$$

Similarly, the CVaR for a trapezoidal fuzzy variable defined by the parameters $\xi = (a_1,\ a_2,\ a_3, a_4)$ and a confidence level $\alpha \in (0,\ 1]$ can be determined using the following equations:

$$\xi_{CVaR}(\alpha) = \begin{cases} \alpha a_1 - (1 + \alpha)a_2 & \alpha \in (0,\ 0.5] \\ (\alpha - 1)a_3 - \alpha a_4 & \alpha \in (0.5,\ 1] \end{cases} \tag{24}$$

In these formulations (Equations 20–24), different expressions for VaR and CVaR are applied depending on whether the confidence level $\alpha$ falls below or above 0.5.

### 3.2.5. Additional constraints *for an* effective and realistic portfolio optimization.

In the realm of real-world portfolio optimization, it is crucial to take into account a variety of practical constraints to ensure that the model genuinely reflects the complexities of actual investment scenarios. These constraints play a vital role in shaping investment strategies that are not only theoretically sound but also feasible and relevant to real-life decision-making. By incorporating practical constraints, we can significantly enhance the realism of the portfolio selection process [75]. For example, constraints related to asset selection, risk exposure, and diversification need help to create a more accurate representation of what investors might face in the market. This alignment with practical investment considerations allows investors to make informed decisions that take into account the limitations and requirements of their specific situations. Furthermore, considering these constraints can lead to better risk management. Constraints such as maximum allowable loss, transaction costs, and liquidity requirements ensure that the portfolio not only seeks returns but also protects

against potential pitfalls. This dual focus on return maximization and risk minimization is essential for developing a robust investment strategy. In addition, practical constraints can facilitate compliance with regulatory requirements and institutional guidelines, especially concerning cryptocurrencies. Many investors, such as pension funds and mutual funds, are subject to rules that dictate how assets can be allocated. By integrating these constraints into the optimization model, we ensure that the resulting portfolios are compliant with relevant regulations, thus avoiding potential legal issues and enhancing investor confidence.

• *Cardinality constraint:* The cardinality constraint imposes a limit on the number of assets that can be included in a portfolio. This limitation is essential for several reasons, primarily for managing transaction costs and fostering effective portfolio diversification. By restricting the number of assets, investors can avoid the complications and expenses associated with trading too many securities, which can erode potential returns. In practical terms, the selection status of each asset within the portfolio is indicated by a binary variable $Z_i$. This means that for each asset, $Z_i$ can either be 0 (indicating that the asset is not included in the portfolio) or 1 (indicating that the asset is included). The cardinality constraint can be mathematically expressed as follows:

$$\sum_{i=1}^{N} Z_i = K \tag{25}$$

Here, $N$ represents the total number of available assets in the market, while $K$ denotes the maximum number of assets that can be included in the portfolio. This equation ensures that the sum of the binary variables does not exceed the specified limit of $K$, enforcing the constraint that only a defined number of assets can be selected.

Additionally, the binary variable $Z_i$ must satisfy the following condition:

$$Z_i \in \{0, 1\}, \quad i = 1, 2, ..., N \tag{26}$$

• *Floor and ceiling constraints:* Floor and ceiling constraints, often referred to as buy-in thresholds, set the minimum and maximum limits for the allocation of portfolio assets. These constraints are critical for maintaining a balanced investment strategy, as they prevent the portfolio from becoming overly concentrated in a single asset and ensure that no asset is allocated an insignificant proportion of the overall investment. By establishing these boundaries, investors can effectively manage their exposure to risk. For instance, a floor constraint ensures that a minimum percentage of the portfolio is invested in a particular asset, while a ceiling constraint caps the maximum percentage that can be allocated. This approach helps to diversify investments and mitigate the potential negative impact of poor performance from any single asset. The mathematical representation of these constraints is expressed as:

$$l_i Z_i \leq x_i \leq u_i Z_i, \quad i = 1, 2, ..., N \tag{27}$$

In this formulation, $x_i$ represents the proportion of the portfolio allocated to asset $i$, while $l_i$ and $u_i$ denote the lower and upper bounds, respectively, for this allocation. The binary variable $Z_i$ indicates whether the asset is included in the portfolio (1) or not (0). Thus, if an asset is not selected (i.e., $Z_i = 0$), the allocation $x_i$ is effectively zero.

Moreover, the following conditions must be satisfied:

$$0 \leq l_i \leq u_i \leq 1 \tag{28}$$

This ensures that the lower bound $l_i$ is non-negative and does not exceed the upper bound $u_i$, which in turn cannot exceed 1 (or 100% of the portfolio).

*Final structure of proposed portfolio optimization model*

Based on all the information provided, this section will present the final structure of the proposed portfolio optimization model in detail. The paper utilizes a credibilistic CVaR approach with trapezoidal fuzzy variables, enhancing its ability to handle uncertainty in asset returns. In addition, the proposed model incorporates cardinality, floor and ceiling constraints to ensure a balanced and realistic investment strategy. Furthermore, the model restricts the confidence level to the interval $\alpha \in (0, \ 0.5]$, which ensures that the assessment of risk remains conservative. This conservative approach is designed to mitigate potential losses and enhance the robustness of the portfolio, leading to more informed investment decisions. The final structure of the proposed portfolio optimization model is established as follows:

$$Min \ CVaR = \sum_{i=1}^{n} x_i \left[ \alpha a_{1i} - (1+\alpha) a_{2i} \right] \tag{29}$$

$$S.t. \quad \sum_{i=1}^{n} x_i \frac{a_{1i} + a_{2i} + a_{3i} + a_{4i}}{4} \geq R \tag{30}$$

$$\sum_{i=1}^{n} x_i = 1 \tag{31}$$

$$l_i Z_i \leq x_i \leq u_i Z_i \tag{32}$$

$$\sum_{i=1}^{n} Z_i = K \tag{33}$$

$$Z_i = \{0, 1\} \tag{34}$$

$$x_i \geq 0, \quad i = 1, \ 2, \ \ldots, \ n \tag{35}$$

In this formulation, Equation (29) specifies the model's objective, which is to minimize the credibilistic CVaR. Equation (30) introduces a constraint related to expected returns. This constraint ensures that the portfolio achieves a minimum required return $R$. By doing so, it establishes a necessary balance between risk and return, allowing investors to pursue profit while managing potential downsides. Next, Equation (31) sets a budget constraint, ensuring that the entire budget is fully allocated across the selected assets. This constraint is vital for maintaining the integrity of the investment strategy, as it prevents any portion of the budget from remaining uninvested. Equation (32) outlines the floor and ceiling constraints, which establish minimum and maximum limits on the allocation for each asset within the portfolio. In this context, $Z_i$ acts as a binary variable that indicates whether a specific asset is included in the portfolio. This mechanism prevents over-concentration in any single asset and ensures that each asset meets its respective allocation thresholds. Additionally, Equation (33) describes the cardinality constraint, which restricts the number of assets included in the portfolio to a maximum of $K$. This limitation reflects practical considerations, such as managing transaction costs and maintaining a manageable portfolio size, thereby enhancing the overall efficiency of the investment strategy. Finally, Equations (34) and

(35) establish essential binary and non-negativity constraints. Equation (34) mandates that $Z_i$ be a binary variable, indicating whether an asset is included in the portfolio (1) or excluded (0). Meanwhile, Equation (35) requires that $x_i$, the proportion allocated to each asset, be non-negative. This condition ensures that short positions are not taken, thereby aligning the model with traditional investment practices that typically avoid negative allocations.

## 4. Numerical experiments

This section is devoted to the numerical experiments and provides a systematic presentation of the results, organized step by step. It is divided into two distinct subsections.

The first subsection outlines the outcomes of the asset preselection process, which employs our novel preselection approach. This segment emphasizes the methodology used as well as the results obtained from selecting assets prior to the optimization phase. The second subsection focuses on the portfolio optimization results derived from the assets identified in the earlier stage, specifically utilizing the credibilistic CVaR model. In this part, we will also explore the validation process conducted to ensure the robustness and effectiveness of the proposed model. Together, these subsections provide a comprehensive overview of the two-stage framework for enhancing cryptocurrency portfolio performance, demonstrating its application and effectiveness in real-world scenarios.

### 4.1. Asset preselection results

As highlighted in Table 2, we collected data on 47 alternative cryptocurrency assets over the period from December 1, 2023, to December 14, 2024. Furthermore, as detailed in Table 3, we established 11 criteria to evaluate the preselection of these crypto assets. These criteria were carefully chosen to reflect key performance indicators and risk factors that are critical for effective portfolio management in the volatile cryptocurrency market. Using the collected data in conjunction with the specified criteria, we developed a decision matrix, which is presented in Table 5. This matrix serves as a vital tool for comparing the assets based on the established criteria, facilitating informed decision-making in the preselection process.

All data used in this analysis were sourced from a variety of reputable databases, ensuring a comprehensive and diverse dataset. It is worth noting that while most criteria can be directly obtained from these databases, two specific metrics — "Avg Active Addresses" and "Circulating/Total Supply" — were not readily available and required further calculation. The methodologies and calculations for these two criteria are detailed in Appendices 14 and 15, respectively.

As noted, we propose a novel approach for the preselection of cryptocurrency assets that begins with calculating results through a diverse set of methods, specifically utilizing 13 MADM techniques outlined in this paper. The techniques employed in our analysis include MARCOS, CODAS, CoCoSo, EDAS, WASPAS, TOPSIS, MOORA, COPRAS, ARAS, VIKOR, MABAC, MACBETH, and TODIM. Once the results are generated from these methods, they are systematically combined using the Copeland approach, which ensures a comprehensive evaluation by aggregating rankings from multiple decision-making techniques.

To implement this framework effectively, we first need to calculate the outputs of all 13 MADM methods using the data provided in the decision matrix. The results of these calculations are systematically organized and presented in Table 6, which provides a clear and comprehensive overview of the findings.

As seen in Table 6, the rankings for each cryptocurrency vary significantly depending on the method used, which can be confusing for investors when selecting the most appropriate approach. For instance, Alternative 33 (ONE, token: Harmony) has the following ranks: 8 in MARCOS, 7 in CODAS, 31 in CoCoSo, 16 in EDAS and WASPAS, 29 in TOPSIS and VIKOR, 2 in MOORA, 32 in COPRAS, 11 in ARAS, 34 in MABAC, 27 in MACBETH, and 18 in TODIM. This wide range of rankings for a single alternative illustrates the inconsistency across different methods, a challenge that extends to other alternatives as well. For more information, the Spearman correlation of these methods is provided in Fig 7. This figure visually represents the correlation coefficients between the various ranking methods, highlighting how closely aligned

 

**Table 5. Decision matrix for cryptocurrency asset preselection.**

| Criteria | | | Avg TPS | Avg Active Addresses | Max Drawdown From ATH | Avg TVL (Billion Dollars) | Avg Fee (Dollars) | Avg Daily Volume (Billion Dollars) | Avg M-Cap (Billion Dollars) | Total Revenue (Million Dollars) | Circu-lating/ Total Supply | Full-Time Devel-opers | Total Devel-opers |
|---|---|---|---|---|---|---|---|---|---|---|---|---|---|
| Alternatives | | Weights | | | | | | | | | | | |
| | | | 11.10% | 9.30% | (3.30%) | 17.80% | (10.90%) | 6.40% | 9.00% | 18.10% | 4.30% | 2.50% | 7.20% |
| $A_1$ | ETH | Ethereum | 22.7 | 6.200 | 94.32% | 21.034 | 4.135 | 20.321 | 250 | 1950.123 | 1.000 | 2788 | 8865 |
| $A_2$ | SOL | Solana | 1053.7 | 6.550 | 97.43% | 0.345 | 0.007 | 1.534 | 50.123 | 100.456 | 0.782 | 664 | 2856 |
| $A_3$ | TRX | Tron | 159 | 13.625 | 96.00% | 5.532 | 0.887 | 1.034 | 80.456 | 40.234 | 0.892 | 135 | 952 |
| $A_4$ | BNB | BSC (Binance) | 378.3 | 11.675 | 85.53% | 2.832 | 0.103 | 1.812 | 350.789 | 800.789 | 0.775 | 556 | 2,015 |
| $A_5$ | BTC | Bitcoin | 6.71 | 12.100 | 84.85% | 0.012 | 15 | 30.543 | 650.987 | 250.345 | 0.905 | 358 | 1,246 |
| $A_6$ | LINK | Chainlink | 7.03 | 0.047 | 90.52% | 0.011 | 0.53 | 0.534 | 20.234 | 20.678 | 0.500 | 61 | 158 |
| $A_7$ | ARB | Arbitrum | 59 | 3.975 | 75.12% | 1.543 | 0.014 | 0.312 | 10.678 | 15.456 | 0.128 | 712 | 2,530 |
| $A_8$ | SUI | Sui | 854.1 | 0.039 | 80.00% | 0.002 | 0.0024 | 0.026 | 5.123 | 5.234 | 0.053 | 202 | 1,108 |
| $A_9$ | AVAX | Avalanche | 89.2 | 0.688 | 92.57% | 0.634 | 0.158 | 0.745 | 15.456 | 30.789 | 0.486 | 496 | 1,706 |
| $A_{10}$ | POL | Polygon | 190.4 | 6.725 | 91.08% | 0.823 | 0.022 | 0.923 | 30.567 | 50.123 | 0.930 | 834 | 2,877 |
| $A_{11}$ | CRV | Curve | 7.56 | 2.150 | 95.68% | 2.345 | 0.503 | 0.035 | 2.345 | 10.345 | 0.227 | 32 | 62 |
| $A_{12}$ | APT | Aptos | 49.5 | 2.892 | 78.49% | 0.008 | 0.003 | 0.423 | 7.456 | 8.234 | 0.200 | 179 | 835 |
| $A_{13}$ | OP | Optimism | 11.8 | 1.600 | 76.82% | 0.657 | 0.0114 | 0.345 | 6.234 | 12.789 | 0.073 | 466 | 1,707 |
| $A_{14}$ | CORE | CORE | 5.49 | 4.225 | 82.64% | 0.004 | 0.5013 | 0.034 | 1.123 | 2.123 | 0.202 | 16 | 41 |
| $A_{15}$ | UNI | Uniswap | 10.95 | 5.808 | 88.03% | 3.067 | 4.88 | 0.645 | 12.345 | 60.456 | 0.753 | 17 | 51 |
| $A_{16}$ | MNT | Mantle | 25.5 | 0.025 | 70.16% | 0.001 | 0.45 | 0.033 | 0.567 | 3.234 | 0.515 | 30 | 130 |
| $A_{17}$ | CRO | Cronos | 72.2 | 6.700 | 89.43% | 0.134 | 0.102 | 0.256 | 3.234 | 25.567 | 0.835 | 16 | 81 |
| $A_{18}$ | ONDO | Ondo | 0.023 | 10.525 | 65.29% | 0.003 | 0.345 | 0.032 | 0.234 | 1.123 | 0.139 | 17 | 39 |
| $A_{19}$ | NUM | Numbers | 0.65 | 0.003 | 68.12% | 0.001 | 0.543 | 0.031 | 0.123 | 0.512 | 0.250 | 12 | 24 |
| $A_{20}$ | CAKE | Pan-cakeswap | 3.067 | 1.306 | 93.55% | 0.045 | 1.86 | 0.03 | 0.345 | 20.678 | 1.000 | 14 | 36 |
| $A_{21}$ | MKR | Maker | 3.69 | 0.316 | 87.43% | 5.034 | 0.693 | 0.145 | 2.456 | 18.345 | 0.972 | 25 | 78 |
| $A_{22}$ | RUNE | Thorchain | 0.021 | 0.007 | 92.43% | 0.245 | 0.123 | 0.134 | 0.789 | 7.234 | 0.660 | 18 | 48 |
| $A_{23}$ | TON | TON | 175 | 3.235 | 74.22% | 0.032 | 0.026 | 0.029 | 0.456 | 40.789 | 0.244 | 52 | 245 |
| $A_{24}$ | ADA | Cardano | 0.084 | 4.010 | 95.07% | 0.215 | 0.153 | 1.245 | 40.234 | 35.123 | 0.778 | 217 | 635 |
| $A_{25}$ | GNO | Gnosis | 65.6 | 0.098 | 80.95% | 0.014 | 0.002 | 0.028 | 0.567 | 5.234 | 0.260 | 257 | 647 |
| $A_{26}$ | AAVE | Aave | 0.275 | 0.121 | 90.46% | 4.567 | 34.76 | 0.212 | 1.789 | 22.789 | 0.875 | 23 | 94 |
| $A_{27}$ | AR | Arweave | 9.34 | 0.168 | 85.44% | 0.009 | 0.397 | 0.027 | 0.234 | 3.123 | 0.758 | 46 | 132 |
| $A_{28}$ | DYDX | dYdX | 1.53 | 0.608 | 88.36% | 0.005 | 0.864 | 0.123 | 0.678 | 10.456 | 0.065 | 30 | 60 |
| $A_{29}$ | NEAR | Near | 117.8 | 2.900 | 91.12% | 0.112 | 0.011 | 0.113 | 0.789 | 15.234 | 0.900 | 322 | 1,214 |
| $A_{30}$ | 1INCH | 1inch | 1.018 | 0.502 | 89.74% | 0.021 | 0.062 | 0.103 | 0.456 | 8.123 | 0.400 | 52 | 122 |
| $A_{31}$ | ROSE | Oasis | 2.83 | 3.889 | 87.53% | 0.003 | 0.627 | 0.052 | 0.234 | 2.345 | 0.570 | 81 | 382 |
| $A_{32}$ | SEI | Sei | 1.664 | 3.945 | 60.29% | 0.002 | 0.488 | 0.051 | 0.123 | 1.678 | 0.180 | 496 | 1,706 |
| $A_{33}$ | ONE | Harmony | 8.941 | 0.041 | 94.44% | 0.011 | 0.0016 | 0.049 | 0.345 | 6.234 | 0.935 | 80 | 228 |
| $A_{34}$ | MANA | Decen-traland | 0.097 | 1.340 | 92.54% | 0.004 | 0.851 | 0.048 | 0.456 | 12.789 | 0.820 | 35 | 50 |
| $A_{35}$ | KAVA | Kava | 0.276 | 1.528 | 86.45% | 0.137 | 0.011 | 0.047 | 0.234 | 4.567 | 0.750 | 28 | 90 |
| $A_{36}$ | RON | Ronin | 18.65 | 2.429 | 72.35% | 0.012 | 0.0031 | 0.046 | 0.123 | 9.345 | 0.150 | 12 | 36 |
| $A_{37}$ | VET | Vechain | 10.25 | 4.001 | 93.50% | 0.008 | 0.056 | 0.045 | 0.345 | 11.678 | 0.838 | 22 | 45 |
| $A_{38}$ | LTC | Litecoin | 3.582 | 3.052 | 90.41% | 0.003 | 0.064 | 0.044 | 50.123 | 20.123 | 0.869 | 33 | 81 |
| $A_{39}$ | EOS | EOS | 3.493 | 1.525 | 95.27% | 0.005 | 0.0124 | 0.043 | 0.789 | 5.456 | 0.980 | 37 | 121 |

*(Continued)*

**Table 5.** (Continued)

| Criteria | | | Avg TPS | Avg Active Addresses | Max Drawdown From ATH | Avg TVL (Billion Dollars) | Avg Fee (Dollars) | Avg Daily Volume (Billion Dollars) | Avg M-Cap (Billion Dollars) | Total Revenue (Million Dollars) | Circu-lating/ Total Supply | Full-Time Devel-opers | Total Devel-opers |
|---|---|---|---|---|---|---|---|---|---|---|---|---|---|
| $A_{40}$ | CELO | Celo | 12.37 | 0.944 | 88.44% | 0.011 | 0.0022 | 0.042 | 0.234 | 3.234 | 0.500 | 342 | 1,206 |
| $A_{41}$ | FTM | Fantom | 59.2 | 0.470 | 91.00% | 0.157 | 0.015 | 0.234 | 0.345 | 7.789 | 0.882 | 269 | 1,013 |
| $A_{42}$ | EGLD | MultiversX | 2.318 | 0.020 | 89.65% | 0.006 | 0.065 | 0.041 | 0.234 | 4.123 | 0.796 | 57 | 181 |
| $A_{43}$ | STX | Stacks | 1.023 | 1.135 | 85.38% | 0.004 | 0.104 | 0.04 | 0.123 | 2.234 | 0.715 | 55 | 155 |
| $A_{44}$ | XMR | Monero | 3.504 | 0.449 | 82.14% | 0.002 | 0.053 | 0.039 | 1.456 | 10.789 | 1.000 | 30 | 80 |
| $A_{45}$ | ATOM | Cosmos | 7.243 | 0.357 | 87.56% | 0.009 | 0.034 | 0.038 | 0.789 | 8.456 | 1.000 | 683 | 2,272 |
| $A_{46}$ | XTZ | Tezos | 2.519 | 1.495 | 90.32% | 0.005 | 0.026 | 0.037 | 0.567 | 6.123 | 1.000 | 72 | 228 |
| $A_{47}$ | ALGO | Algorand | 170.3 | 1.273 | 92.18% | 0.007 | 0.0019 | 0.036 | 0.234 | 5.789 | 0.780 | 87 | 444 |

the rankings are across different approaches. A higher correlation indicates a greater agreement in rankings, while lower values suggest significant discrepancies.

As illustrated in the analysis, there is a significant variation among the various ranking methods employed. This discrepancy highlights the necessity for a robust framework that can effectively synthesize these differing outcomes. Therefore, implementing our proposed method becomes crucial in ensuring a reliable evaluation process.

To address this challenge, we have integrated the results from the 13 selected methods to derive a single comprehensive ranking. This was achieved through three distinct combined approaches: (a) Mean Rank, (b) Borda Count, and (c) Copeland. The outcomes from the Mean Rank method are detailed in Table 7, providing a foundational overview. Following this, Table 8 presents a pairwise matrix of 47 cryptocurrency assets based on the results outlined in Table 6. Table 9 then showcases the results from both the Borda Count and Copeland methods, offering further insights into the rankings derived from these approaches. This integrated methodology aims to provide investors with a clearer and more consistent evaluation framework, enhancing decision-making processes in the ever-evolving landscape of investment opportunities. By reconciling the differences among the various methods, we strive to present a unified perspective that better informs investment strategies.

In Table 7, we calculated the mean rank of each asset, providing a foundational metric to assess their performance relative to one another. This mean rank serves as a critical indicator of each asset's standing, with lower values reflecting better rankings.

Subsequently, we constructed a Pairwise Comparison Matrix in Table 8. This matrix serves as a detailed framework for evaluating the relative performance of each asset by comparing them in pairs. For each asset pair, we assessed their standings based on the number of better ranks they received across the 13 selected methods. In this comparison, the asset that received a greater number of better ranks was designated as a win (1). Conversely, the asset with fewer better ranks was classified as a loss (0). For instance, in this paper, consider two assets: if one asset ranked better than the other in at least 7 of the 13 methods, it would be classified as a win (1). On the other hand, if the same asset had a lower count of better ranks compared to its counterpart, it would be classified as a loss (0).

After completing Table 8, the Pairwise Comparison Matrix, we analyzed the results to derive meaningful insights. The sum of each row corresponds to the total wins for that specific asset, reflecting the number of other assets it has outperformed. Meanwhile, the sum of each column indicates the total losses, showcasing how many assets surpassed it in rank. This dual perspective provides a comprehensive view of each asset's competitive standing. With this data in hand, we can proceed to calculate the Borda Count and Copeland scores. The Borda Count method uses the total number of wins for each asset; thus, an asset with a higher total win's count is assigned a better rank. In contrast, the Copeland score

is derived from the difference between wins and losses—each asset is ranked more favorably if it has a higher positive difference. This approach highlights not only the wins but also the impact of losses on the asset's overall standing. Finally, Table 9 presents the results of the Borda Count and Copeland methods, offering deeper insights into the rankings derived from this comprehensive analysis.

Based on the results presented in Table 8, we constructed the Pairwise Comparison Matrix, enabling us to calculate the total wins and losses for each asset. This matrix serves as a foundational tool for evaluating asset performance, allowing for a systematic comparison. Utilizing this data, we derived the rankings based on both the Borda count and Copeland methods, which are summarized in Table 9. Importantly, in this study, the rankings from both methods are identical due to the use of an odd number of comparison methods (13 methods).

When the number of methods is odd, the outcomes are limited to two possibilities: a win (1) or a loss (0). A win occurs when an asset ranks higher than its counterpart in at least 7 of the 13 methods, while a loss is assigned if it ranks lower. In this context, it becomes evident that an asset with a higher number of wins will correspondingly have fewer losses, resulting in a significant difference between the wins and losses. Consequently, the Borda count, which focuses solely on the tally of wins, aligns perfectly with Copeland's method, which takes into account the overall difference between wins and losses. Conversely, the scenario shifts when the number of methods is even. In such cases, we encounter three potential outcomes: win (1), loss (0), and draw (0). For example, if there are 12 methods and asset $i$ ranks better than asset $j$ in 6 methods, while asset $j$ ranks better than asset $i$ in the remaining 6, this situation results in a draw. This complicates our ability to ascertain that an asset with a greater number of wins necessarily has fewer losses, as the presence of draws introduces ambiguity into the comparisons. As a result, we would expect the Borda count to yield different outcomes compared to Copeland's method in this context. Given these considerations, Copeland's approach emerges as the more robust method for consolidating results, as it effectively accounts for both wins and losses. Although our findings reveal that the results from the Borda count and Copeland are consistent in this study (attributable to the 13 methods utilized), we aimed to present a comprehensive framework in this paper. This framework serves as a benchmark for readers who may explore various scenarios in future analyses. It is crucial to emphasize that if the number of methods changes, particularly to an even count, the effectiveness of the Borda count for making accurate comparisons may be diminished. Additionally, the summarized results from the Mean Rank, Borda Count, and Copeland methods are presented in Table A3 (Appendix 16). This table illustrates that while both the Borda count and Copeland methods may yield similar rankings under certain conditions, they can differ significantly from the mean rank approach, underscoring the importance of selecting an appropriate method based on the specific context of the analysis.

As previously noted, we employed the Copeland method for our comparisons. The results derived from this approach are displayed in Table 10, where we have categorized the assets into five distinct groups: excellent, satisfactory, average, below average, and poor. Each category is visually differentiated by a specific color in the table, enhancing clarity and facilitating quick assessment.

The categorization of assets into distinct groups enhances clarity and provides a clear guide for asset evaluation. Each category—excellent, satisfactory, average, below average, and poor—has its own economic significance. The "excellent" category represents the top-performing assets, indicating their strong potential for high returns and stability, making them ideal candidates for inclusion in a portfolio focused on maximizing performance. On the other hand, assets categorized as "poor" highlight those with lower expected returns or higher associated risks, serving as a signal for investors to avoid or limit exposure. By utilizing these categories, investors can align their portfolios more effectively with their financial goals, risk preferences, and market conditions.

In the subsequent steps of our analysis, we will focus on the assets classified as excellent (Top 10 assets), as this category contains high-quality assets with strong potential for superior performance. By concentrating on these high-quality assets, we have effectively conducted a preselection of the leading cryptocurrency options, ensuring that only the most promising candidates are considered for our investment strategy. This targeted focus on high-quality assets allows for a

**Table 6. Cryptocurrency rankings according to different methods.**

| Alternative | MARCOS | | CODAS | | CoCoSo ($\lambda = 0.5$) | | EDAS | | WASPAS | | TOPSIS | | MOORA | |
|---|---|---|---|---|---|---|---|---|---|---|---|---|---|---|
| | Result | Rank | Result | Rank | Result | Rank | Result | Rank | Result | Rank | Result | Rank | Result | Rank |
| $A_1$ | 0.6423 | 1 | 10.331 | 1 | 4.5515 | 1 | 0.5463 | 1 | 0.2843 | 1 | 0.7318 | 1 | 0.1659 | 1 |
| $A_2$ | 0.2863 | 4 | 3.179 | 4 | 2.7170 | 5 | 0.1205 | 28 | 0.1489 | 3 | 0.3488 | 3 | 0.0985 | 9 |
| $A_3$ | 0.2419 | 5 | 2.146 | 5 | 2.7578 | 4 | 0.0870 | 43 | 0.1040 | 4 | 0.3258 | 4 | 0.0960 | 43 |
| $A_4$ | 0.3336 | 2 | 3.643 | 3 | 3.2140 | 2 | 0.1770 | 4 | 0.1621 | 2 | 0.4048 | 2 | 0.0982 | 26 |
| $A_5$ | 0.3264 | 3 | 3.822 | 2 | 2.8093 | 3 | 0.3930 | 3 | 0.0611 | 6 | 0.3210 | 5 | 0.0742 | 46 |
| $A_6$ | 0.0527 | 44 | -5.065 | 44 | 1.7489 | 30 | 0.1114 | 33 | 0.0138 | 36 | 0.2640 | 39 | 0.0970 | 37 |
| $A_7$ | 0.1009 | 21 | -2.229 | 21 | 2.0252 | 11 | 0.0776 | 44 | 0.0424 | 8 | 0.2765 | 9 | 0.0985 | 14 |
| $A_8$ | 0.1932 | 6 | 1.917 | 6 | 1.7400 | 34 | 0.1698 | 5 | 0.0321 | 12 | 0.3110 | 6 | 0.0985 | 6 |
| $A_9$ | 0.0740 | 33 | -3.912 | 35 | 1.8323 | 20 | 0.0753 | 45 | 0.0306 | 14 | 0.2708 | 17 | 0.0980 | 30 |
| $A_{10}$ | 0.1611 | 9 | -0.350 | 11 | 2.3237 | 6 | 0.0559 | 47 | 0.0681 | 5 | 0.2834 | 8 | 0.0984 | 16 |
| $A_{11}$ | 0.0684 | 36 | -4.072 | 38 | 1.8244 | 22 | 0.1033 | 37 | 0.0227 | 22 | 0.2747 | 11 | 0.0971 | 36 |
| $A_{12}$ | 0.1282 | 13 | -0.941 | 13 | 1.9145 | 15 | 0.0971 | 39 | 0.0345 | 11 | 0.2697 | 18 | 0.0985 | 7 |
| $A_{13}$ | 0.0691 | 35 | -4.238 | 41 | 1.7413 | 33 | 0.1039 | 36 | 0.0259 | 20 | 0.2692 | 20 | 0.0985 | 12 |
| $A_{14}$ | 0.0635 | 39 | -4.112 | 39 | 1.7552 | 29 | 0.1258 | 21 | 0.0117 | 38 | 0.2653 | 36 | 0.0971 | 35 |
| $A_{15}$ | 0.1311 | 12 | -0.893 | 12 | 2.1538 | 7 | 0.1140 | 30 | 0.0426 | 7 | 0.2615 | 44 | 0.0847 | 45 |
| $A_{16}$ | 0.0556 | 42 | -4.838 | 43 | 1.7099 | 37 | 0.1283 | 15 | 0.0097 | 45 | 0.2634 | 41 | 0.0972 | 33 |
| $A_{17}$ | 0.1185 | 14 | -1.202 | 15 | 2.1078 | 8 | 0.0943 | 41 | 0.0362 | 9 | 0.2739 | 12 | 0.0982 | 25 |
| $A_{18}$ | 0.1095 | 18 | -0.942 | 14 | 1.9337 | 13 | 0.1385 | 7 | 0.0165 | 31 | 0.2753 | 10 | 0.0975 | 31 |
| $A_{19}$ | 0.0408 | 46 | -5.972 | 46 | 1.2719 | 47 | 0.1419 | 6 | 0.0062 | 47 | 0.2620 | 42 | 0.0970 | 38 |
| $A_{20}$ | 0.0765 | 31 | -3.068 | 30 | 1.7827 | 27 | 0.1235 | 25 | 0.0152 | 33 | 0.2568 | 46 | 0.0932 | 44 |
| $A_{21}$ | 0.1132 | 16 | -1.312 | 16 | 2.0743 | 9 | 0.1253 | 22 | 0.0306 | 15 | 0.2934 | 7 | 0.0965 | 40 |
| $A_{22}$ | 0.0551 | 43 | -4.664 | 42 | 1.6217 | 43 | 0.1279 | 17 | 0.0106 | 44 | 0.2658 | 32 | 0.0981 | 28 |
| $A_{23}$ | 0.0912 | 23 | -2.751 | 26 | 1.9675 | 12 | 0.0923 | 42 | 0.0281 | 19 | 0.2733 | 13 | 0.0984 | 17 |
| $A_{24}$ | 0.1033 | 19 | -1.933 | 20 | 2.0305 | 10 | 0.0697 | 46 | 0.0311 | 13 | 0.2720 | 14 | 0.0981 | 29 |
| $A_{25}$ | 0.1389 | 10 | 0.220 | 9 | 1.7462 | 32 | 0.1121 | 31 | 0.0282 | 18 | 0.2678 | 23 | 0.0985 | 4 |
| $A_{26}$ | 0.1030 | 20 | -1.791 | 19 | 1.3864 | 46 | 0.5274 | 2 | 0.0211 | 25 | 0.1162 | 47 | 0.0348 | 47 |
| $A_{27}$ | 0.0604 | 41 | -4.218 | 40 | 1.7207 | 36 | 0.1301 | 13 | 0.0113 | 42 | 0.2638 | 40 | 0.0974 | 32 |
| $A_{28}$ | 0.0320 | 47 | -6.738 | 47 | 1.5845 | 44 | 0.1330 | 8 | 0.0073 | 46 | 0.2608 | 45 | 0.0960 | 42 |
| $A_{29}$ | 0.1143 | 15 | -1.393 | 18 | 1.9071 | 16 | 0.0963 | 40 | 0.0301 | 17 | 0.2709 | 16 | 0.0985 | 10 |
| $A_{30}$ | 0.0484 | 45 | -5.451 | 45 | 1.6790 | 40 | 0.1310 | 11 | 0.0115 | 41 | 0.2655 | 35 | 0.0983 | 22 |
| $A_{31}$ | 0.0786 | 30 | -3.215 | 32 | 1.8259 | 21 | 0.1098 | 34 | 0.0138 | 35 | 0.2650 | 38 | 0.0967 | 39 |
| $A_{32}$ | 0.0730 | 34 | -3.640 | 33 | 1.6726 | 41 | 0.1261 | 19 | 0.0116 | 40 | 0.2652 | 37 | 0.0971 | 34 |
| $A_{33}$ | 0.1750 | 8 | 1.756 | 7 | 1.7466 | 31 | 0.1282 | 16 | 0.0305 | 16 | 0.2664 | 29 | 0.0985 | 2 |
| $A_{34}$ | 0.0683 | 37 | -3.709 | 34 | 1.7351 | 35 | 0.1240 | 23 | 0.0117 | 37 | 0.2619 | 43 | 0.0961 | 41 |
| $A_{35}$ | 0.0843 | 28 | -2.923 | 29 | 1.7907 | 26 | 0.1235 | 24 | 0.0179 | 28 | 0.2668 | 27 | 0.0985 | 10 |
| $A_{36}$ | 0.1103 | 17 | -1.369 | 17 | 1.5765 | 45 | 0.1234 | 26 | 0.0224 | 23 | 0.2670 | 24 | 0.0985 | 8 |
| $A_{37}$ | 0.0907 | 24 | -2.414 | 23 | 1.8726 | 18 | 0.1183 | 29 | 0.0186 | 27 | 0.2685 | 21 | 0.0983 | 21 |
| $A_{38}$ | 0.0932 | 22 | -2.352 | 22 | 1.9336 | 14 | 0.1010 | 38 | 0.0208 | 26 | 0.2693 | 19 | 0.0983 | 23 |
| $A_{39}$ | 0.0900 | 25 | -2.444 | 24 | 1.8027 | 25 | 0.1261 | 20 | 0.0179 | 29 | 0.2668 | 26 | 0.0985 | 13 |
| $A_{40}$ | 0.1347 | 11 | -0.143 | 10 | 1.6288 | 42 | 0.1286 | 14 | 0.0236 | 21 | 0.2661 | 31 | 0.0985 | 5 |
| $A_{41}$ | 0.0859 | 26 | -2.763 | 27 | 1.7698 | 28 | 0.1118 | 32 | 0.0217 | 24 | 0.2678 | 22 | 0.0985 | 15 |
| $A_{42}$ | 0.0620 | 40 | -4.066 | 37 | 1.6969 | 38 | 0.1318 | 10 | 0.0112 | 43 | 0.2656 | 34 | 0.0983 | 24 |
| $A_{43}$ | 0.0657 | 38 | -4.013 | 36 | 1.6830 | 39 | 0.1273 | 18 | 0.0116 | 39 | 0.2656 | 33 | 0.0982 | 27 |
| $A_{44}$ | 0.0762 | 32 | -3.085 | 31 | 1.8210 | 23 | 0.1304 | 12 | 0.0146 | 34 | 0.2662 | 30 | 0.0983 | 20 |
| $A_{45}$ | 0.0813 | 29 | -2.875 | 28 | 1.8106 | 24 | 0.1320 | 9 | 0.0156 | 32 | 0.2666 | 28 | 0.0984 | 19 |
| $A_{46}$ | 0.0855 | 27 | -2.657 | 25 | 1.8354 | 19 | 0.1221 | 27 | 0.0169 | 30 | 0.2669 | 25 | 0.0984 | 17 |
| $A_{47}$ | 0.1787 | 7 | 1.100 | 8 | 1.9009 | 17 | 0.1062 | 35 | 0.0354 | 10 | 0.2712 | 15 | 0.0985 | 3 |

Source: Authors' own computation

| COPRAS | | ARAS | | VIKOR ($\vartheta = 0.5$) | | MABAC | | MACBETH | | TODIM ($\theta = 1$) | |
|--------|------|--------|------|--------|------|--------|------|--------|------|--------|------|
| Result | Rank | Result | Rank | Result | Rank | Result | Rank | Result | Rank | Result | Rank |
| 0.2537 | 1 | 1 | 1 | 1.0000 | 1 | 0.535 | 1 | 66.468 | 1 | 1 | 1 |
| 0.0644 | 4 | 0.2560 | 4 | 0.4615 | 3 | 0.164 | 4 | 27.342 | 4 | 0.7580 | 3 |
| 0.0482 | 5 | 0.1869 | 5 | 0.4280 | 4 | 0.142 | 5 | 25.516 | 5 | 0.6277 | 5 |
| 0.0962 | 2 | 0.3688 | 2 | 0.5149 | 2 | 0.242 | 2 | 33.079 | 3 | 0.7607 | 2 |
| 0.0908 | 3 | 0.3597 | 3 | 0.4086 | 5 | 0.190 | 3 | 37.161 | 2 | 0.6757 | 4 |
| 0.0064 | 36 | 0.0204 | 35 | 0.2854 | 45 | -0.043 | 44 | 5.743 | 38 | 0.2081 | 33 |
| 0.0183 | 11 | 0.0637 | 15 | 0.3274 | 13 | 0.003 | 14 | 7.283 | 28 | 0.4181 | 9 |
| 0.0315 | 6 | 0.1522 | 6 | 0.3506 | 7 | 0.032 | 7 | 11.069 | 12 | 0.2609 | 25 |
| 0.0134 | 16 | 0.0442 | 23 | 0.3056 | 27 | -0.023 | 28 | 7.914 | 23 | 0.3456 | 15 |
| 0.0255 | 7 | 0.0925 | 8 | 0.3724 | 6 | 0.060 | 6 | 15.841 | 7 | 0.6275 | 6 |
| 0.0134 | 18 | 0.0477 | 20 | 0.2949 | 38 | -0.032 | 37 | 7.820 | 24 | 0.1307 | 42 |
| 0.0134 | 17 | 0.0721 | 14 | 0.3117 | 22 | -0.017 | 24 | 5.875 | 36 | 0.3965 | 11 |
| 0.0110 | 21 | 0.0370 | 24 | 0.2992 | 31 | -0.033 | 38 | 4.010 | 42 | 0.2991 | 22 |
| 0.0058 | 39 | 0.0178 | 36 | 0.2976 | 34 | -0.028 | 31 | 5.834 | 37 | 0.1030 | 45 |
| 0.0213 | 9 | 0.0852 | 10 | 0.2906 | 41 | 0.024 | 10 | 14.761 | 8 | 0.3647 | 12 |
| 0.0047 | 42 | 0.0128 | 43 | 0.2980 | 33 | -0.028 | 32 | 3.556 | 44 | 0.1475 | 40 |
| 0.0139 | 14 | 0.0450 | 22 | 0.3433 | 8 | 0.024 | 9 | 12.021 | 9 | 0.3539 | 14 |
| 0.0105 | 22 | 0.0341 | 26 | 0.3430 | 9 | 0.027 | 8 | 8.180 | 20 | 0.1214 | 43 |
| 0.0027 | 47 | 0.0059 | 47 | 0.2854 | 44 | -0.043 | 43 | 1.788 | 47 | 0.0000 | 47 |
| 0.0047 | 43 | 0.0177 | 37 | 0.2858 | 43 | -0.024 | 29 | 9.027 | 17 | 0.1756 | 37 |
| 0.0230 | 8 | 0.0869 | 9 | 0.3335 | 10 | 0.020 | 11 | 11.622 | 10 | 0.2974 | 23 |
| 0.0058 | 40 | 0.0137 | 41 | 0.2912 | 40 | -0.042 | 42 | 6.003 | 34 | 0.1042 | 44 |
| 0.0158 | 13 | 0.0508 | 18 | 0.3247 | 14 | -0.001 | 15 | 6.809 | 29 | 0.3075 | 20 |
| 0.0159 | 12 | 0.0539 | 17 | 0.3280 | 12 | 0.005 | 12 | 11.190 | 11 | 0.4398 | 7 |
| 0.0105 | 23 | 0.0783 | 13 | 0.2962 | 36 | -0.037 | 40 | 4.343 | 41 | 0.3328 | 16 |
| 0.0197 | 10 | 0.0798 | 12 | 0.0000 | 47 | -0.099 | 47 | 21.636 | 6 | 0.2542 | 27 |
| 0.0046 | 44 | 0.0123 | 45 | 0.2962 | 35 | -0.031 | 36 | 5.946 | 35 | 0.1790 | 36 |
| 0.0030 | 46 | 0.0089 | 46 | 0.2649 | 46 | -0.065 | 46 | 3.439 | 46 | 0.0339 | 46 |
| 0.0129 | 19 | 0.0470 | 21 | 0.3292 | 11 | 0.005 | 13 | 10.354 | 13 | 0.4246 | 8 |
| 0.0058 | 41 | 0.0128 | 44 | 0.2866 | 42 | -0.048 | 45 | 4.810 | 40 | 0.1381 | 41 |
| 0.0072 | 31 | 0.0243 | 31 | 0.3062 | 26 | -0.016 | 23 | 8.052 | 21 | 0.2445 | 29 |
| 0.0067 | 35 | 0.0210 | 34 | 0.3139 | 19 | -0.008 | 19 | 3.894 | 43 | 0.2307 | 32 |
| 0.0071 | 32 | 0.0801 | 11 | 0.3020 | 29 | -0.030 | 34 | 7.482 | 27 | 0.3315 | 18 |
| 0.0046 | 45 | 0.0149 | 39 | 0.2930 | 39 | -0.029 | 33 | 7.722 | 26 | 0.1746 | 38 |
| 0.0075 | 29 | 0.0254 | 28 | 0.3076 | 25 | -0.022 | 27 | 6.788 | 31 | 0.2363 | 31 |
| 0.0082 | 26 | 0.0493 | 19 | 0.3015 | 30 | -0.030 | 35 | 3.492 | 45 | 0.1727 | 39 |
| 0.0087 | 25 | 0.0247 | 30 | 0.3187 | 17 | -0.008 | 18 | 9.540 | 14 | 0.2379 | 30 |
| 0.0113 | 20 | 0.0346 | 25 | 0.3225 | 15 | -0.003 | 16 | 9.492 | 15 | 0.3113 | 19 |
| 0.0073 | 30 | 0.0247 | 29 | 0.3094 | 24 | -0.020 | 26 | 8.587 | 18 | 0.2593 | 26 |
| 0.0070 | 33 | 0.0610 | 16 | 0.2959 | 37 | -0.037 | 41 | 5.645 | 39 | 0.2924 | 24 |
| 0.0092 | 24 | 0.0301 | 27 | 0.3101 | 23 | -0.019 | 25 | 7.925 | 22 | 0.3326 | 17 |
| 0.0060 | 38 | 0.0136 | 42 | 0.2982 | 32 | -0.034 | 39 | 6.274 | 32 | 0.1931 | 35 |
| 0.0063 | 37 | 0.0146 | 40 | 0.3033 | 28 | -0.027 | 30 | 6.242 | 33 | 0.1991 | 34 |
| 0.0069 | 34 | 0.0163 | 38 | 0.3131 | 21 | -0.015 | 22 | 6.803 | 30 | 0.2483 | 28 |
| 0.0080 | 27 | 0.0216 | 33 | 0.3133 | 20 | -0.015 | 21 | 7.756 | 25 | 0.3576 | 13 |
| 0.0079 | 28 | 0.0224 | 32 | 0.3141 | 18 | -0.014 | 20 | 8.332 | 19 | 0.3021 | 21 |
| 0.0134 | 15 | 0.0946 | 7 | 0.3195 | 16 | -0.007 | 17 | 9.290 | 16 | 0.4030 | 10 |

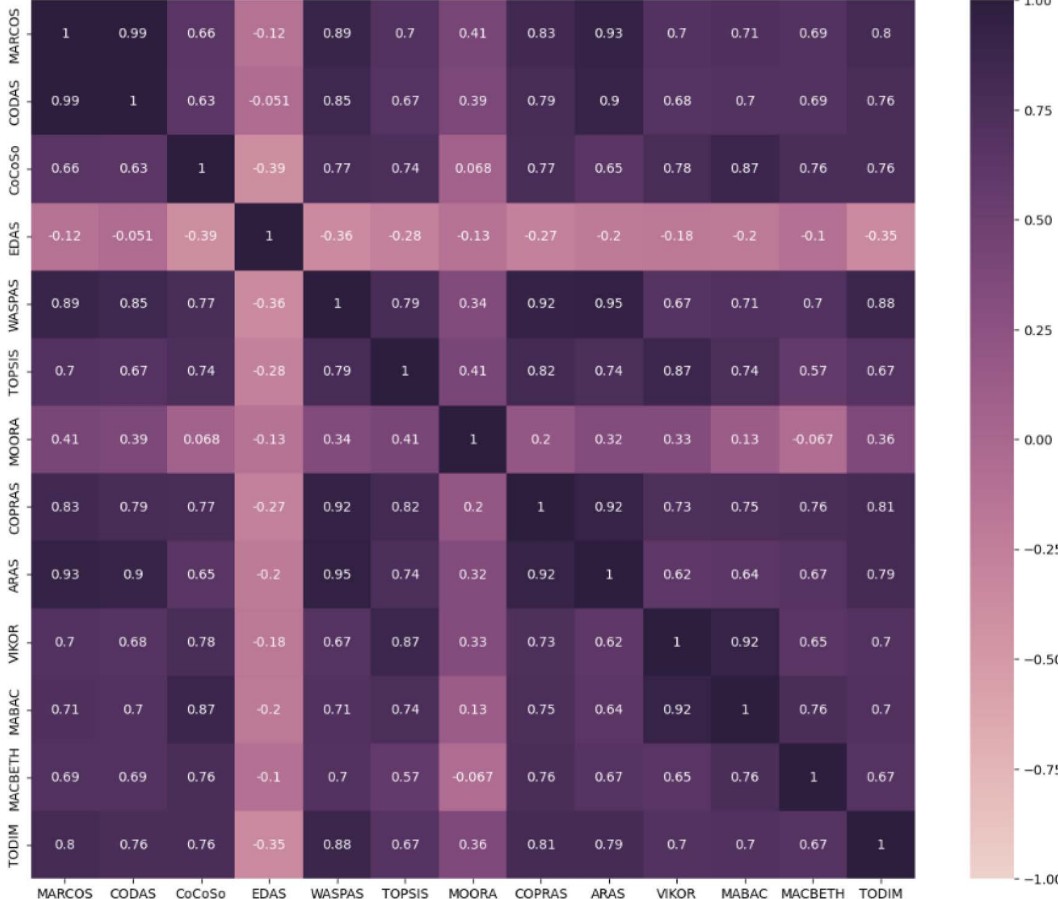

**Fig 7. Spearman correlation coefficients among 13 selected MADM methods.**

more refined and informed investment approach, ensuring that the portfolio remains aligned with the investor's risk-return objectives.

### 4.2. Portfolio optimization results

Following the preselection of assets, which outlines the results of Stage 1, this section focuses on the portfolio optimization process, specifically regarding asset weight allocation. In this stage, we will allocate weights to the assets classified as excellent, specifically the top 10 assets, under various scenarios. These scenarios will be constructed by varying the values assigned to the cardinality constraints, enabling us to create portfolios of different sizes tailored to the preferences and risk appetites of diverse investors. To verify the efficiency of our proposed model and validate our two-stage framework, we will evaluate the portfolios generated using the top 10 assets against those developed using larger groups, including the top 20, 30, 40, and all 47 assets. For each of these portfolios, we will establish scenarios analogous to our own results, ensuring that the comparisons are meaningful and relevant. This validation process will provide insights into the effectiveness of our preselection methodology and its impact on portfolio performance. In this section, we will detail the results step by step, illustrating the implications of our findings.

As previously highlighted, we employ the Credibilistic CVaR criterion, which incorporates practical constraints into our analysis. To accurately model this Credibilistic CVaR, we utilize trapezoidal fuzzy variables. The initial step in this process

**Table 7. Combined results of 13 methods using the mean rank approach.**

| | MAR-COS | CODAS | CoCoSo ($\lambda = 0.5$) | EDAS | WASPAS | TOP-SIS | MOORA | COPRAS | ARAS | VIKOR ($\vartheta = 0.5$) | MABAC | MAC-BETH | TODIM ($\theta = 1$) | Mean Rank | Final Rank |
|---|---|---|---|---|---|---|---|---|---|---|---|---|---|---|---|
| $A_1$ | 1 | 1 | 1 | 1 | 1 | 1 | 1 | 1 | 1 | 1 | 1 | 1 | 1 | 1.00 | 1 |
| $A_2$ | 4 | 4 | 5 | 28 | 3 | 3 | 9 | 4 | 4 | 3 | 4 | 4 | 3 | 6.00 | 3 |
| $A_3$ | 5 | 5 | 4 | 43 | 4 | 4 | 43 | 5 | 5 | 4 | 5 | 5 | 5 | 10.54 | 5 |
| $A_4$ | 2 | 3 | 2 | 4 | 2 | 2 | 26 | 2 | 2 | 2 | 2 | 3 | 2 | 4.15 | 2 |
| $A_5$ | 3 | 2 | 3 | 3 | 6 | 5 | 46 | 3 | 3 | 5 | 3 | 2 | 4 | 6.77 | 4 |
| $A_6$ | 44 | 44 | 30 | 33 | 36 | 39 | 37 | 36 | 35 | 45 | 44 | 38 | 33 | 38.00 | 45 |
| $A_7$ | 21 | 21 | 11 | 44 | 8 | 9 | 14 | 11 | 15 | 13 | 14 | 28 | 9 | 16.77 | 12 |
| $A_8$ | 6 | 6 | 34 | 5 | 12 | 6 | 6 | 6 | 6 | 7 | 7 | 12 | 25 | 10.62 | 6 |
| $A_9$ | 33 | 35 | 20 | 45 | 14 | 17 | 30 | 16 | 23 | 27 | 28 | 23 | 15 | 25.08 | 27 |
| $A_{10}$ | 9 | 11 | 6 | 47 | 5 | 8 | 16 | 7 | 8 | 6 | 6 | 7 | 6 | 10.92 | 7 |
| $A_{11}$ | 36 | 38 | 22 | 37 | 22 | 11 | 36 | 18 | 20 | 38 | 37 | 24 | 42 | 29.31 | 33 |
| $A_{12}$ | 13 | 13 | 15 | 39 | 11 | 18 | 7 | 17 | 14 | 22 | 24 | 36 | 11 | 18.46 | 14 |
| $A_{13}$ | 35 | 41 | 33 | 36 | 20 | 20 | 12 | 21 | 24 | 31 | 38 | 42 | 22 | 28.85 | 32 |
| $A_{14}$ | 39 | 39 | 29 | 21 | 38 | 36 | 35 | 39 | 36 | 34 | 31 | 37 | 45 | 35.31 | 39 |
| $A_{15}$ | 12 | 12 | 7 | 30 | 7 | 44 | 45 | 9 | 10 | 41 | 10 | 8 | 12 | 19.00 | 15 |
| $A_{16}$ | 42 | 43 | 37 | 15 | 45 | 41 | 33 | 42 | 43 | 33 | 32 | 44 | 40 | 37.69 | 42 |
| $A_{17}$ | 14 | 15 | 8 | 41 | 9 | 12 | 25 | 14 | 22 | 8 | 9 | 9 | 14 | 15.38 | 10 |
| $A_{18}$ | 18 | 14 | 13 | 7 | 31 | 10 | 31 | 22 | 26 | 9 | 8 | 20 | 43 | 19.38 | 16 |
| $A_{19}$ | 46 | 46 | 47 | 6 | 47 | 42 | 38 | 47 | 47 | 44 | 43 | 47 | 47 | 42.08 | 46 |
| $A_{20}$ | 31 | 30 | 27 | 25 | 33 | 46 | 44 | 43 | 37 | 43 | 29 | 17 | 37 | 34.00 | 37 |
| $A_{21}$ | 16 | 16 | 9 | 22 | 15 | 7 | 40 | 8 | 9 | 10 | 11 | 10 | 23 | 15.08 | 9 |
| $A_{22}$ | 43 | 42 | 43 | 17 | 44 | 32 | 28 | 40 | 41 | 40 | 42 | 34 | 44 | 37.69 | 42 |
| $A_{23}$ | 23 | 26 | 12 | 42 | 19 | 13 | 17 | 13 | 18 | 14 | 15 | 29 | 20 | 20.08 | 18 |
| $A_{24}$ | 19 | 20 | 10 | 46 | 13 | 14 | 29 | 12 | 17 | 12 | 12 | 11 | 7 | 17.08 | 13 |
| $A_{25}$ | 10 | 9 | 32 | 31 | 18 | 23 | 4 | 23 | 13 | 36 | 40 | 41 | 16 | 22.77 | 20 |
| $A_{26}$ | 20 | 19 | 46 | 2 | 25 | 47 | 47 | 10 | 12 | 47 | 47 | 6 | 27 | 27.31 | 30 |
| $A_{27}$ | 41 | 40 | 36 | 13 | 42 | 40 | 32 | 44 | 45 | 35 | 36 | 35 | 36 | 36.54 | 41 |
| $A_{28}$ | 47 | 47 | 44 | 8 | 46 | 45 | 42 | 46 | 46 | 46 | 46 | 46 | 46 | 42.69 | 47 |
| $A_{29}$ | 15 | 18 | 16 | 40 | 17 | 16 | 10 | 19 | 21 | 11 | 13 | 13 | 8 | 16.69 | 11 |
| $A_{30}$ | 45 | 45 | 40 | 11 | 41 | 35 | 22 | 41 | 44 | 42 | 45 | 40 | 41 | 37.85 | 44 |
| $A_{31}$ | 30 | 32 | 21 | 34 | 35 | 38 | 39 | 31 | 31 | 26 | 23 | 21 | 29 | 30.00 | 34 |
| $A_{32}$ | 34 | 33 | 41 | 19 | 40 | 37 | 34 | 35 | 34 | 19 | 19 | 43 | 32 | 32.31 | 35 |
| $A_{33}$ | 8 | 7 | 31 | 16 | 16 | 29 | 2 | 32 | 11 | 29 | 34 | 27 | 18 | 20.00 | 17 |
| $A_{34}$ | 37 | 34 | 35 | 23 | 37 | 43 | 41 | 45 | 39 | 39 | 33 | 26 | 38 | 36.15 | 40 |
| $A_{35}$ | 28 | 29 | 26 | 24 | 28 | 27 | 10 | 29 | 28 | 25 | 27 | 31 | 31 | 26.38 | 28 |
| $A_{36}$ | 17 | 17 | 45 | 26 | 23 | 24 | 8 | 26 | 19 | 30 | 35 | 45 | 39 | 27.23 | 29 |
| $A_{37}$ | 24 | 23 | 18 | 29 | 27 | 21 | 21 | 25 | 30 | 17 | 18 | 14 | 30 | 22.85 | 21 |
| $A_{38}$ | 22 | 22 | 14 | 38 | 26 | 19 | 23 | 20 | 25 | 15 | 16 | 15 | 19 | 21.08 | 19 |
| $A_{39}$ | 25 | 24 | 25 | 20 | 29 | 26 | 13 | 30 | 29 | 24 | 26 | 18 | 26 | 24.23 | 25 |
| $A_{40}$ | 11 | 10 | 42 | 14 | 21 | 31 | 5 | 33 | 16 | 37 | 41 | 39 | 24 | 24.92 | 26 |
| $A_{41}$ | 26 | 27 | 28 | 32 | 24 | 22 | 15 | 24 | 27 | 23 | 25 | 22 | 17 | 24.00 | 24 |
| $A_{42}$ | 40 | 37 | 38 | 10 | 43 | 34 | 24 | 38 | 42 | 32 | 39 | 32 | 35 | 34.15 | 38 |
| $A_{43}$ | 38 | 36 | 39 | 18 | 39 | 33 | 27 | 37 | 40 | 28 | 30 | 33 | 34 | 33.23 | 36 |
| $A_{44}$ | 32 | 31 | 23 | 12 | 34 | 30 | 20 | 34 | 38 | 21 | 22 | 30 | 28 | 27.31 | 30 |
| $A_{45}$ | 29 | 28 | 24 | 9 | 32 | 28 | 19 | 27 | 33 | 20 | 21 | 25 | 13 | 23.69 | 22 |
| $A_{46}$ | 27 | 25 | 19 | 27 | 30 | 25 | 17 | 28 | 32 | 18 | 20 | 19 | 21 | 23.69 | 22 |
| $A_{47}$ | 7 | 8 | 17 | 35 | 10 | 15 | 3 | 15 | 7 | 16 | 17 | 16 | 10 | 13.54 | 8 |

Source: Authors' own computation

**Table 8. Pairwise comparison matrix of 47 cryptocurrency assets.**

| | 1 | 2 | 3 | 4 | 5 | 6 | 7 | 8 | 9 | 10 | 11 | 12 | 13 | 14 | 15 | 16 | 17 | 18 | 19 | 20 | 21 | 22 | 23 | 24 | 25 |
|---|---|---|---|---|---|---|---|---|---|---|---|---|---|---|---|---|---|---|---|---|---|---|---|---|---|
| 1 | - | 1 | 1 | 1 | 1 | 1 | 1 | 1 | 1 | 1 | 1 | 1 | 1 | 1 | 1 | 1 | 1 | 1 | 1 | 1 | 1 | 1 | 1 | 1 | 1 |
| 2 | 0 | - | 1 | 0 | 0 | 1 | 1 | 1 | 1 | 1 | 1 | 1 | 1 | 1 | 1 | 1 | 1 | 1 | 1 | 1 | 1 | 1 | 1 | 1 | 1 |
| 3 | 0 | 0 | - | 0 | 0 | 1 | 1 | 1 | 1 | 1 | 1 | 1 | 1 | 1 | 1 | 1 | 1 | 1 | 1 | 1 | 1 | 1 | 1 | 1 | 1 |
| 4 | 0 | 1 | 1 | - | 1 | 1 | 1 | 1 | 1 | 1 | 1 | 1 | 1 | 1 | 1 | 1 | 1 | 1 | 1 | 1 | 1 | 1 | 1 | 1 | 1 |
| 5 | 0 | 1 | 1 | 0 | - | 1 | 1 | 1 | 1 | 1 | 1 | 1 | 1 | 1 | 1 | 1 | 1 | 1 | 1 | 1 | 1 | 1 | 1 | 1 | 1 |
| 6 | 0 | 0 | 0 | 0 | 0 | - | 0 | 0 | 0 | 0 | 0 | 0 | 0 | 0 | 0 | 1 | 0 | 0 | 1 | 0 | 0 | 0 | 0 | 0 | 0 |
| 7 | 0 | 0 | 0 | 0 | 0 | 1 | - | 0 | 1 | 0 | 1 | 1 | 1 | 1 | 0 | 1 | 0 | 1 | 1 | 1 | 0 | 1 | 1 | 0 | 1 |
| 8 | 0 | 0 | 0 | 0 | 0 | 1 | 1 | - | 1 | 1 | 1 | 1 | 1 | 1 | 1 | 1 | 1 | 1 | 1 | 1 | 1 | 1 | 1 | 1 | 1 |
| 9 | 0 | 0 | 0 | 0 | 0 | 1 | 0 | 0 | - | 0 | 1 | 0 | 1 | 1 | 0 | 1 | 0 | 0 | 1 | 1 | 0 | 1 | 0 | 0 | 1 |
| 10 | 0 | 0 | 0 | 0 | 0 | 1 | 1 | 0 | 1 | - | 1 | 1 | 1 | 1 | 1 | 1 | 1 | 1 | 1 | 1 | 1 | 1 | 1 | 1 | 1 |
| 11 | 0 | 0 | 0 | 0 | 0 | 1 | 0 | 0 | 0 | 0 | - | 0 | 1 | 1 | 0 | 1 | 0 | 0 | 1 | 1 | 0 | 1 | 0 | 0 | 0 |
| 12 | 0 | 0 | 0 | 0 | 0 | 1 | 0 | 0 | 1 | 0 | 1 | - | 1 | 1 | 0 | 1 | 0 | 1 | 1 | 1 | 0 | 1 | 1 | 0 | 1 |
| 13 | 0 | 0 | 0 | 0 | 0 | 1 | 0 | 0 | 0 | 0 | 0 | 0 | - | 1 | 0 | 1 | 0 | 0 | 1 | 1 | 0 | 1 | 0 | 0 | 0 |
| 14 | 0 | 0 | 0 | 0 | 0 | 1 | 0 | 0 | 0 | 0 | 0 | 0 | 0 | - | 0 | 1 | 0 | 0 | 1 | 0 | 0 | 1 | 0 | 0 | 0 |
| 15 | 0 | 0 | 0 | 0 | 0 | 1 | 1 | 0 | 1 | 0 | 1 | 1 | 1 | 1 | - | 1 | 1 | 1 | 1 | 1 | 1 | 1 | 1 | 1 | 1 |
| 16 | 0 | 0 | 0 | 0 | 0 | 0 | 0 | 0 | 0 | 0 | 0 | 0 | 0 | 0 | 0 | - | 0 | 0 | 1 | 0 | 0 | 0 | 0 | 0 | 0 |
| 17 | 0 | 0 | 0 | 0 | 0 | 1 | 1 | 0 | 1 | 0 | 1 | 1 | 1 | 1 | 0 | 1 | - | 1 | 1 | 1 | 1 | 1 | 1 | 1 | 1 |
| 18 | 0 | 0 | 0 | 0 | 0 | 1 | 0 | 0 | 1 | 0 | 1 | 0 | 1 | 1 | 0 | 1 | 0 | - | 1 | 1 | 0 | 1 | 1 | 0 | 1 |
| 19 | 0 | 0 | 0 | 0 | 0 | 0 | 0 | 0 | 0 | 0 | 0 | 0 | 0 | 0 | 0 | 0 | 0 | 0 | - | 0 | 0 | 0 | 0 | 0 | 0 |
| 20 | 0 | 0 | 0 | 0 | 0 | 1 | 0 | 0 | 0 | 0 | 0 | 0 | 0 | 1 | 0 | 1 | 0 | 0 | 1 | - | 0 | 1 | 0 | 0 | 0 |
| 21 | 0 | 0 | 0 | 0 | 0 | 1 | 1 | 0 | 1 | 0 | 1 | 1 | 1 | 1 | 0 | 1 | 0 | 1 | 1 | 1 | - | 1 | 1 | 1 | 1 |
| 22 | 0 | 0 | 0 | 0 | 0 | 1 | 0 | 0 | 0 | 0 | 0 | 0 | 0 | 0 | 0 | 1 | 0 | 0 | 1 | 0 | 0 | - | 0 | 0 | 0 |
| 23 | 0 | 0 | 0 | 0 | 0 | 1 | 0 | 0 | 1 | 0 | 1 | 0 | 1 | 1 | 0 | 1 | 0 | 0 | 1 | 1 | 0 | 1 | - | 0 | 0 |
| 24 | 0 | 0 | 0 | 0 | 0 | 1 | 1 | 0 | 1 | 0 | 1 | 1 | 1 | 1 | 0 | 1 | 0 | 1 | 1 | 1 | 0 | 1 | 1 | - | 1 |
| 25 | 0 | 0 | 0 | 0 | 0 | 1 | 0 | 0 | 0 | 0 | 1 | 0 | 1 | 1 | 0 | 1 | 0 | 0 | 1 | 1 | 0 | 1 | 1 | 0 | - |
| 26 | 0 | 0 | 0 | 0 | 0 | 1 | 0 | 0 | 0 | 0 | 1 | 0 | 0 | 1 | 0 | 1 | 0 | 0 | 1 | 1 | 0 | 1 | 0 | 0 | 0 |
| 27 | 0 | 0 | 0 | 0 | 0 | 1 | 0 | 0 | 0 | 0 | 0 | 0 | 0 | 0 | 0 | 1 | 0 | 0 | 1 | 0 | 0 | 1 | 0 | 0 | 0 |
| 28 | 0 | 0 | 0 | 0 | 0 | 0 | 0 | 0 | 0 | 0 | 0 | 0 | 0 | 0 | 0 | 0 | 0 | 0 | 0 | 0 | 0 | 0 | 0 | 0 | 0 |
| 29 | 0 | 0 | 0 | 0 | 0 | 1 | 1 | 0 | 1 | 0 | 1 | 0 | 1 | 1 | 0 | 1 | 0 | 1 | 1 | 1 | 0 | 1 | 1 | 0 | 1 |
| 30 | 0 | 0 | 0 | 0 | 0 | 0 | 0 | 0 | 0 | 0 | 0 | 0 | 0 | 0 | 0 | 0 | 0 | 0 | 1 | 0 | 0 | 0 | 0 | 0 | 0 |
| 31 | 0 | 0 | 0 | 0 | 0 | 1 | 0 | 0 | 0 | 0 | 1 | 0 | 1 | 1 | 0 | 1 | 0 | 0 | 1 | 1 | 0 | 1 | 0 | 0 | 0 |
| 32 | 0 | 0 | 0 | 0 | 0 | 1 | 0 | 0 | 0 | 0 | 1 | 0 | 0 | 1 | 0 | 1 | 0 | 0 | 1 | 1 | 0 | 1 | 0 | 0 | 0 |
| 33 | 0 | 0 | 0 | 0 | 0 | 1 | 0 | 0 | 0 | 0 | 1 | 0 | 1 | 1 | 0 | 1 | 0 | 0 | 1 | 1 | 0 | 1 | 1 | 0 | 1 |
| 34 | 0 | 0 | 0 | 0 | 0 | 0 | 0 | 0 | 0 | 0 | 0 | 0 | 0 | 0 | 0 | 1 | 0 | 0 | 1 | 0 | 0 | 1 | 0 | 0 | 0 |
| 35 | 0 | 0 | 0 | 0 | 0 | 1 | 0 | 0 | 0 | 0 | 1 | 0 | 1 | 1 | 0 | 1 | 0 | 0 | 1 | 1 | 0 | 1 | 0 | 0 | 0 |
| 36 | 0 | 0 | 0 | 0 | 0 | 1 | 0 | 0 | 0 | 0 | 1 | 0 | 1 | 1 | 0 | 1 | 0 | 0 | 1 | 1 | 0 | 1 | 0 | 0 | 0 |
| 37 | 0 | 0 | 0 | 0 | 0 | 1 | 0 | 0 | 1 | 0 | 1 | 0 | 1 | 1 | 0 | 1 | 0 | 0 | 1 | 1 | 0 | 1 | 0 | 0 | 0 |
| 38 | 0 | 0 | 0 | 0 | 0 | 1 | 0 | 0 | 1 | 0 | 1 | 0 | 1 | 1 | 0 | 1 | 0 | 0 | 1 | 1 | 0 | 1 | 0 | 0 | 0 |
| 39 | 0 | 0 | 0 | 0 | 0 | 1 | 0 | 0 | 1 | 0 | 1 | 0 | 1 | 1 | 0 | 1 | 0 | 0 | 1 | 1 | 0 | 1 | 0 | 0 | 0 |
| 40 | 0 | 0 | 0 | 0 | 0 | 1 | 0 | 0 | 0 | 0 | 1 | 0 | 0 | 1 | 0 | 1 | 0 | 0 | 1 | 1 | 0 | 1 | 0 | 0 | 0 |
| 41 | 0 | 0 | 0 | 0 | 0 | 1 | 0 | 0 | 1 | 0 | 1 | 0 | 1 | 1 | 0 | 1 | 0 | 0 | 1 | 1 | 0 | 1 | 0 | 0 | 0 |
| 42 | 0 | 0 | 0 | 0 | 0 | 1 | 0 | 0 | 0 | 0 | 0 | 0 | 0 | 0 | 0 | 1 | 0 | 0 | 1 | 0 | 0 | 0 | 0 | 0 | 0 |
| 43 | 0 | 0 | 0 | 0 | 0 | 1 | 0 | 0 | 0 | 0 | 0 | 0 | 0 | 0 | 0 | 1 | 0 | 0 | 1 | 0 | 0 | 1 | 0 | 0 | 0 |
| 44 | 0 | 0 | 0 | 0 | 0 | 1 | 0 | 0 | 0 | 0 | 1 | 0 | 1 | 1 | 0 | 1 | 0 | 0 | 1 | 1 | 0 | 1 | 0 | 0 | 0 |
| 45 | 0 | 0 | 0 | 0 | 0 | 1 | 0 | 0 | 1 | 0 | 1 | 0 | 1 | 1 | 0 | 1 | 0 | 0 | 1 | 1 | 0 | 1 | 0 | 0 | 0 |
| 46 | 0 | 0 | 0 | 0 | 0 | 1 | 0 | 0 | 1 | 0 | 1 | 0 | 1 | 1 | 0 | 1 | 0 | 0 | 1 | 1 | 0 | 1 | 0 | 0 | 0 |
| 47 | 0 | 0 | 0 | 0 | 0 | 1 | 0 | 0 | 1 | 0 | 1 | 1 | 1 | 1 | 1 | 1 | 0 | 1 | 1 | 1 | 0 | 1 | 1 | 0 | 1 |

Source: Authors' own computation

| 26 | 27 | 28 | 29 | 30 | 31 | 32 | 33 | 34 | 35 | 36 | 37 | 38 | 39 | 40 | 41 | 42 | 43 | 44 | 45 | 46 | 47 |
|----|----|----|----|----|----|----|----|----|----|----|----|----|----|----|----|----|----|----|----|----|----|
| 1 | 1 | 1 | 1 | 1 | 1 | 1 | 1 | 1 | 1 | 1 | 1 | 1 | 1 | 1 | 1 | 1 | 1 | 1 | 1 | 1 | 1 |
| 1 | 1 | 1 | 1 | 1 | 1 | 1 | 1 | 1 | 1 | 1 | 1 | 1 | 1 | 1 | 1 | 1 | 1 | 1 | 1 | 1 | 1 |
| 1 | 1 | 1 | 1 | 1 | 1 | 1 | 1 | 1 | 1 | 1 | 1 | 1 | 1 | 1 | 1 | 1 | 1 | 1 | 1 | 1 | 1 |
| 1 | 1 | 1 | 1 | 1 | 1 | 1 | 1 | 1 | 1 | 1 | 1 | 1 | 1 | 1 | 1 | 1 | 1 | 1 | 1 | 1 | 1 |
| 1 | 1 | 1 | 1 | 1 | 1 | 1 | 1 | 1 | 1 | 1 | 1 | 1 | 1 | 1 | 1 | 1 | 1 | 1 | 1 | 1 | 1 |
| 0 | 0 | 1 | 0 | 1 | 0 | 0 | 0 | 1 | 0 | 0 | 0 | 0 | 0 | 0 | 0 | 0 | 0 | 0 | 0 | 0 | 0 |
| 1 | 1 | 1 | 0 | 1 | 1 | 1 | 1 | 1 | 1 | 1 | 1 | 1 | 1 | 1 | 1 | 1 | 1 | 1 | 1 | 1 | 1 |
| 1 | 1 | 1 | 1 | 1 | 1 | 1 | 1 | 1 | 1 | 1 | 1 | 1 | 1 | 1 | 1 | 1 | 1 | 1 | 1 | 1 | 1 |
| 1 | 1 | 1 | 0 | 1 | 1 | 1 | 1 | 1 | 1 | 1 | 0 | 0 | 0 | 1 | 0 | 1 | 1 | 1 | 0 | 0 | 0 |
| 1 | 1 | 1 | 1 | 1 | 1 | 1 | 1 | 1 | 1 | 1 | 1 | 1 | 1 | 1 | 1 | 1 | 1 | 1 | 1 | 1 | 1 |
| 0 | 1 | 1 | 0 | 1 | 0 | 0 | 0 | 1 | 0 | 0 | 0 | 0 | 0 | 0 | 0 | 1 | 1 | 0 | 0 | 0 | 0 |
| 1 | 1 | 1 | 1 | 1 | 1 | 1 | 1 | 1 | 1 | 1 | 1 | 1 | 1 | 1 | 1 | 1 | 1 | 1 | 1 | 1 | 0 |
| 1 | 1 | 1 | 0 | 1 | 0 | 1 | 0 | 1 | 0 | 0 | 0 | 0 | 0 | 1 | 0 | 1 | 1 | 0 | 0 | 0 | 0 |
| 0 | 1 | 1 | 0 | 1 | 0 | 0 | 0 | 1 | 0 | 0 | 0 | 0 | 0 | 0 | 0 | 0 | 0 | 0 | 0 | 0 | 0 |
| 1 | 1 | 1 | 1 | 1 | 1 | 1 | 1 | 1 | 1 | 1 | 1 | 1 | 1 | 1 | 1 | 1 | 1 | 1 | 1 | 1 | 0 |
| 0 | 0 | 1 | 0 | 1 | 0 | 0 | 0 | 0 | 0 | 0 | 0 | 0 | 0 | 0 | 0 | 0 | 0 | 0 | 0 | 0 | 0 |
| 1 | 1 | 1 | 1 | 1 | 1 | 1 | 1 | 1 | 1 | 1 | 1 | 1 | 1 | 1 | 1 | 1 | 1 | 1 | 1 | 1 | 1 |
| 1 | 1 | 1 | 0 | 1 | 1 | 1 | 1 | 1 | 1 | 1 | 1 | 1 | 1 | 1 | 1 | 1 | 1 | 1 | 1 | 1 | 0 |
| 0 | 0 | 1 | 0 | 0 | 0 | 0 | 0 | 0 | 0 | 0 | 0 | 0 | 0 | 0 | 0 | 0 | 0 | 0 | 0 | 0 | 0 |
| 0 | 1 | 1 | 0 | 1 | 0 | 0 | 0 | 1 | 0 | 0 | 0 | 0 | 0 | 0 | 0 | 1 | 1 | 0 | 0 | 0 | 0 |
| 1 | 1 | 1 | 1 | 1 | 1 | 1 | 1 | 1 | 1 | 1 | 1 | 1 | 1 | 1 | 1 | 1 | 1 | 1 | 1 | 1 | 1 |
| 0 | 0 | 1 | 0 | 1 | 0 | 0 | 0 | 0 | 0 | 0 | 0 | 0 | 0 | 0 | 0 | 0 | 0 | 0 | 0 | 0 | 0 |
| 1 | 1 | 1 | 0 | 1 | 1 | 1 | 0 | 1 | 1 | 1 | 1 | 1 | 1 | 1 | 1 | 1 | 1 | 1 | 1 | 1 | 0 |
| 1 | 1 | 1 | 1 | 1 | 1 | 1 | 1 | 1 | 1 | 1 | 1 | 1 | 1 | 1 | 1 | 1 | 1 | 1 | 1 | 1 | 1 |
| 1 | 1 | 1 | 0 | 1 | 1 | 1 | 0 | 1 | 1 | 1 | 1 | 1 | 1 | 1 | 1 | 1 | 1 | 1 | 1 | 1 | 0 |
| - | 1 | 1 | 0 | 1 | 1 | 1 | 0 | 1 | 1 | 0 | 1 | 1 | 1 | 0 | 0 | 1 | 1 | 1 | 1 | 1 | 0 |
| 0 | - | 1 | 0 | 1 | 0 | 0 | 0 | 0 | 0 | 0 | 0 | 0 | 0 | 0 | 0 | 0 | 0 | 0 | 0 | 0 | 0 |
| 0 | 0 | - | 0 | 0 | 0 | 0 | 0 | 0 | 0 | 0 | 0 | 0 | 0 | 0 | 0 | 0 | 0 | 0 | 0 | 0 | 0 |
| 1 | 1 | 1 | - | 1 | 1 | 1 | 1 | 1 | 1 | 1 | 1 | 1 | 1 | 1 | 1 | 1 | 1 | 1 | 1 | 1 | 0 |
| 0 | 0 | 1 | 0 | - | 0 | 0 | 0 | 0 | 0 | 0 | 0 | 0 | 0 | 0 | 0 | 0 | 0 | 0 | 0 | 0 | 0 |
| 0 | 1 | 1 | 0 | 1 | - | 1 | 0 | 1 | 0 | 0 | 0 | 0 | 0 | 0 | 0 | 1 | 1 | 0 | 0 | 0 | 0 |
| 0 | 1 | 1 | 0 | 1 | 0 | - | 0 | 1 | 0 | 0 | 0 | 0 | 0 | 0 | 0 | 1 | 1 | 0 | 0 | 0 | 0 |
| 1 | 1 | 1 | 0 | 1 | 1 | 1 | - | 1 | 1 | 1 | 1 | 1 | 1 | 1 | 0 | 1 | 1 | 1 | 0 | 1 | 0 |
| 0 | 1 | 1 | 0 | 1 | 0 | 0 | 0 | - | 0 | 0 | 0 | 0 | 0 | 0 | 0 | 1 | 0 | 0 | 0 | 0 | 0 |
| 0 | 1 | 1 | 0 | 1 | 1 | 1 | 0 | 1 | - | 0 | 0 | 0 | 0 | 0 | 0 | 1 | 1 | 1 | 0 | 0 | 0 |
| 1 | 1 | 1 | 0 | 1 | 1 | 1 | 0 | 1 | 1 | - | 0 | 0 | 1 | 0 | 0 | 1 | 1 | 1 | 1 | 1 | 0 |
| 0 | 1 | 1 | 0 | 1 | 1 | 1 | 0 | 1 | 1 | 1 | - | 0 | 1 | 0 | 1 | 1 | 1 | 1 | 1 | 1 | 0 |
| 0 | 1 | 1 | 0 | 1 | 1 | 1 | 0 | 1 | 1 | 1 | 1 | - | 1 | 1 | 1 | 1 | 1 | 1 | 1 | 1 | 0 |
| 0 | 1 | 1 | 0 | 1 | 1 | 1 | 0 | 1 | 1 | 0 | 0 | 0 | - | 0 | 0 | 1 | 1 | 1 | 1 | 1 | 0 |
| 1 | 1 | 1 | 0 | 1 | 1 | 1 | 0 | 1 | 1 | 1 | 1 | 0 | 1 | - | 0 | 1 | 1 | 1 | 0 | 0 | 0 |
| 1 | 1 | 1 | 0 | 1 | 1 | 1 | 1 | 1 | 1 | 1 | 0 | 0 | 1 | 1 | - | 1 | 1 | 1 | 1 | 1 | 0 |
| 0 | 1 | 1 | 0 | 1 | 0 | 0 | 0 | 0 | 0 | 0 | 0 | 0 | 0 | 0 | 0 | - | 0 | 0 | 0 | 0 | 0 |
| 0 | 1 | 1 | 0 | 1 | 0 | 0 | 0 | 1 | 0 | 0 | 0 | 0 | 0 | 0 | 0 | 1 | - | 0 | 0 | 0 | 0 |
| 0 | 1 | 1 | 0 | 1 | 1 | 1 | 0 | 1 | 0 | 0 | 0 | 0 | 0 | 0 | 0 | 1 | 1 | - | 0 | 0 | 0 |
| 0 | 1 | 1 | 0 | 1 | 1 | 1 | 1 | 1 | 1 | 0 | 0 | 0 | 0 | 1 | 0 | 1 | 1 | 1 | - | 0 | 0 |
| 0 | 1 | 1 | 0 | 1 | 1 | 1 | 0 | 1 | 1 | 0 | 0 | 0 | 0 | 1 | 0 | 1 | 1 | 1 | 1 | - | 0 |
| 1 | 1 | 1 | 1 | 1 | 1 | 1 | 1 | 1 | 1 | 1 | 1 | 1 | 1 | 1 | 1 | 1 | 1 | 1 | 1 | 1 | - |

**Table 9. Combined results of 13 methods using the Borda count and Copeland approaches.**

| Alternative | | | Total Wins | Total Losses | Difference between Wins and Losses | Borda Count | Copeland |
|---|---|---|---|---|---|---|---|
| $A_1$ | ETH | Ethereum | 46 | 0 | 46 | 1 | 1 |
| $A_2$ | SOL | Solana | 43 | 3 | 40 | 4 | 4 |
| $A_3$ | TRX | Tron | 42 | 4 | 38 | 5 | 5 |
| $A_4$ | BNB | BSC (Binance) | 45 | 1 | 44 | 2 | 2 |
| $A_5$ | BTC | Bitcoin | 44 | 2 | 42 | 3 | 3 |
| $A_6$ | LINK | Chainlink | 5 | 41 | -36 | 42 | 42 |
| $A_7$ | ARB | Arbitrum | 34 | 12 | 22 | 13 | 13 |
| $A_8$ | SUI | Sui | 41 | 5 | 36 | 6 | 6 |
| $A_9$ | AVAX | Avalanche | 23 | 23 | 0 | 23 | 23 |
| $A_{10}$ | POL | Polygon | 40 | 6 | 34 | 7 | 7 |
| $A_{11}$ | CRV | Curve | 13 | 33 | -20 | 34 | 34 |
| $A_{12}$ | APT | Aptos | 33 | 13 | 20 | 14 | 14 |
| $A_{13}$ | OP | Optimism | 15 | 31 | -16 | 32 | 32 |
| $A_{14}$ | CORE | CORE | 8 | 38 | -30 | 38 | 38 |
| $A_{15}$ | UNI | Uniswap | 38 | 8 | 30 | 8 | 8 |
| $A_{16}$ | MNT | Mantle | 3 | 43 | -40 | 44 | 44 |
| $A_{17}$ | CRO | Cronos | 38 | 8 | 30 | 8 | 8 |
| $A_{18}$ | ONDO | Ondo | 31 | 15 | 16 | 16 | 16 |
| $A_{19}$ | NUM | Numbers | 1 | 45 | -44 | 46 | 46 |
| $A_{20}$ | CAKE | Pancakeswap | 11 | 35 | -24 | 36 | 36 |
| $A_{21}$ | MKR | Maker | 37 | 9 | 28 | 10 | 10 |
| $A_{22}$ | RUNE | Thorchain | 5 | 41 | -36 | 42 | 42 |
| $A_{23}$ | TON | TON | 28 | 18 | 10 | 17 | 17 |
| $A_{24}$ | ADA | Cardano | 36 | 10 | 26 | 11 | 11 |
| $A_{25}$ | GNO | Gnosis | 28 | 18 | 10 | 17 | 17 |
| $A_{26}$ | AAVE | Aave | 22 | 24 | -2 | 24 | 24 |
| $A_{27}$ | AR | Arweave | 6 | 40 | -34 | 41 | 41 |
| $A_{28}$ | DYDX | dYdX | 0 | 46 | -46 | 47 | 47 |
| $A_{29}$ | NEAR | Near | 33 | 13 | 20 | 14 | 14 |
| $A_{30}$ | 1INCH | 1inch | 2 | 44 | -42 | 45 | 45 |
| $A_{31}$ | ROSE | Oasis | 15 | 31 | -16 | 32 | 32 |
| $A_{32}$ | SEI | Sei | 13 | 33 | -20 | 34 | 34 |
| $A_{33}$ | ONE | Harmony | 27 | 19 | 8 | 19 | 19 |
| $A_{34}$ | MANA | Decentraland | 7 | 39 | -32 | 40 | 40 |
| $A_{35}$ | KAVA | Kava | 17 | 29 | -12 | 30 | 30 |
| $A_{36}$ | RON | Ronin | 22 | 24 | -2 | 24 | 24 |
| $A_{37}$ | VET | Vechain | 24 | 22 | 2 | 22 | 22 |
| $A_{38}$ | LTC | Litecoin | 26 | 20 | 6 | 20 | 20 |
| $A_{39}$ | EOS | EOS | 21 | 25 | -4 | 26 | 26 |
| $A_{40}$ | CELO | Celo | 21 | 25 | -4 | 26 | 26 |
| $A_{41}$ | FTM | Fantom | 26 | 20 | 6 | 20 | 20 |
| $A_{42}$ | EGLD | MultiversX | 8 | 38 | -30 | 38 | 38 |
| $A_{43}$ | STX | Stacks | 10 | 36 | -26 | 37 | 37 |
| $A_{44}$ | XMR | Monero | 16 | 30 | -14 | 31 | 31 |
| $A_{45}$ | ATOM | Cosmos | 21 | 25 | -4 | 26 | 26 |
| $A_{46}$ | XTZ | Tezos | 21 | 25 | -4 | 26 | 26 |
| $A_{47}$ | ALGO | Algorand | 35 | 11 | 24 | 12 | 12 |

Source: Authors' own computation

Table 10. Categorization of cryptocurrency assets based on Copeland method results.

| Alternative | | | Final Rank | Alternative | | | Final Rank |
|---|---|---|---|---|---|---|---|
| $A_1$ | ETH | Ethereum | 1 | $A_{36}$ | RON | Ronin | 24 |
| $A_4$ | BNB | BSC (Binance) | 2 | $A_{39}$ | EOS | EOS | 26 |
| $A_5$ | BTC | Bitcoin | 3 | $A_{40}$ | CELO | Celo | 26 |
| $A_2$ | SOL | Solana | 4 | $A_{45}$ | ATOM | Cosmos | 26 |
| $A_3$ | TRX | Tron | 5 | $A_{46}$ | XTZ | Tezos | 26 |
| $A_8$ | SUI | Sui | 6 | $A_{35}$ | KAVA | Kava | 30 |
| $A_{10}$ | POL | Polygon | 7 | $A_{44}$ | XMR | Monero | 31 |
| $A_{15}$ | UNI | Uniswap | 8 | $A_{13}$ | OP | Optimism | 32 |
| $A_{17}$ | CRO | Cronos | 8 | $A_{31}$ | ROSE | Oasis | 32 |
| $A_{21}$ | MKR | Maker | 10 | $A_{11}$ | CRV | Curve | 34 |
| $A_{24}$ | ADA | Cardano | 11 | $A_{32}$ | SEI | Sei | 34 |
| $A_{47}$ | ALGO | Algorand | 12 | $A_{20}$ | CAKE | Pancakeswap | 36 |
| $A_7$ | ARB | Arbitrum | 13 | $A_{43}$ | STX | Stacks | 37 |
| $A_{12}$ | APT | Aptos | 14 | $A_{14}$ | CORE | CORE | 38 |
| $A_{29}$ | NEAR | Near | 14 | $A_{42}$ | EGLD | MultiversX | 38 |
| $A_{18}$ | ONDO | Ondo | 16 | $A_{34}$ | MANA | Decentraland | 40 |
| $A_{23}$ | TON | TON | 17 | $A_{27}$ | AR | Arweave | 41 |
| $A_{25}$ | GNO | Gnosis | 17 | $A_6$ | LINK | Chainlink | 42 |
| $A_{33}$ | ONE | Harmony | 19 | $A_{22}$ | RUNE | Thorchain | 42 |
| $A_{38}$ | LTC | Litecoin | 20 | $A_{16}$ | MNT | Mantle | 44 |
| $A_{41}$ | FTM | Fantom | 20 | $A_{30}$ | 1INCH | 1inch | 45 |
| $A_{37}$ | VET | Vechain | 22 | $A_{19}$ | NUM | Numbers | 46 |
| $A_9$ | AVAX | Avalanche | 23 | $A_{28}$ | DYDX | dYdX | 47 |
| $A_{26}$ | AAVE | Aave | 24 | | | | |

Source: Authors' own computation

involves gathering trapezoidal fuzzy variables for each asset, which are derived from expert opinions and compiled in Table 11. This data will serve as a foundational element for our analysis, allowing us to assess the potential risks and returns associated with the selected portfolios.

For the optimization process, we utilized the data presented in Table 11, along with the Credibilistic CVaR framework defined in Equations 29–35. We employed GAMS software to solve the optimization model, carefully setting the parameters as follows: $\alpha = 0.05$, an expected return ($R$) of 10%, a floor ($l_i$) of 0.1, and a ceiling ($u_i$) of 0.5. The cardinality constraints were established for $K$ values of 3, 5, and 7, allowing us to explore different portfolio sizes. After solving the model, we constructed three distinct portfolios using the top 10 assets, each corresponding to the specified cardinality constraints. For instance, when $K = 3$, the selected asset allocations were: $A_3$ (TRX) at 50% (ranked 5), $A_{15}$ (UNI) at 10% (ranked 8), and $A_{17}$ (CRO) at 40% (ranked 9). These allocations reflect a strategic selection aimed at optimizing the risk-return profile within the set of constraints. The detailed results of these portfolios, including the asset weights and corresponding ranks, are summarized in Table 12. This comprehensive analysis allows us to assess how different cardinality constraints influence portfolio composition and performance.

To verify the efficiency of our proposed model and validate our two-stage framework, we expanded the scope of our analysis by generating additional portfolios using larger asset groups. Specifically, we examined the top 20, 30, 40, and all 47 assets, allowing for a comprehensive evaluation across multiple scenarios. The results of these portfolio constructions

**Table 11. Trapezoidal fuzzy representation of cryptocurrency returns expectations.**

| Rank | Alternative | | Trapezoidal Fuzzy Data | Rank | Alternative | | Trapezoidal Fuzzy Data |
|---|---|---|---|---|---|---|---|
| $x_1$ | $A_1$ | ETH | (-0.101, -0.028, 0.118, 0.19) | $x_{25}$ | $A_{36}$ | RON | (-0.181, -0.093, 0.082, 0.17) |
| $x_2$ | $A_4$ | BNB | (-0.085, -0.023, 0.103, 0.165) | $x_{26}$ | $A_{39}$ | EOS | (-0.216, -0.102, 0.125, 0.239) |
| $x_3$ | $A_5$ | BTC | (-0.082, -0.031, 0.071, 0.123) | $x_{27}$ | $A_{40}$ | CELO | (-0.189, -0.016, 0.33, 0.503) |
| $x_4$ | $A_2$ | SOL | (-0.136, -0.066, 0.074, 0.144) | $x_{28}$ | $A_{45}$ | ATOM | (-0.169, -0.08, 0.099, 0.189) |
| $x_5$ | $A_3$ | TRX | (-0.209, 0.065, 0.614, 0.889) | $x_{29}$ | $A_{46}$ | XTZ | (-0.182, -0.015, 0.32, 0.488) |
| $x_6$ | $A_8$ | SUI | (-0.166, -0.026, 0.253, 0.393) | $x_{30}$ | $A_{35}$ | KAVA | (-0.204, -0.109, 0.082, 0.178) |
| $x_7$ | $A_{10}$ | POL | (-0.17, -0.089, 0.073, 0.154) | $x_{31}$ | $A_{44}$ | XMR | (-0.366, -0.215, 0.087, 0.238) |
| $x_8$ | $A_{15}$ | UNI | (-0.141, 0.029, 0.368, 0.538) | $x_{32}$ | $A_{13}$ | OP | (-0.165, -0.025, 0.254, 0.394) |
| $x_9$ | $A_{17}$ | CRO | (-0.148, 0.056, 0.465, 0.67) | $x_{33}$ | $A_{31}$ | ROSE | (-0.195, -0.1, 0.091, 0.186) |
| $x_{10}$ | $A_{21}$ | MKR | (-0.145, -0.05, 0.141, 0.237) | $x_{34}$ | $A_{11}$ | CRV | (-0.204, -0.084, 0.154, 0.273) |
| $x_{11}$ | $A_{24}$ | ADA | (-0.159, -0.062, 0.131, 0.228) | $x_{35}$ | $A_{32}$ | SEI | (-0.182, -0.069, 0.158, 0.271) |
| $x_{12}$ | $A_{47}$ | ALGO | (-0.159, -0.025, 0.242, 0.376) | $x_{36}$ | $A_{20}$ | CAKE | (-0.197, -0.095, 0.109, 0.21) |
| $x_{13}$ | $A_7$ | ARB | (-0.171, -0.067, 0.14, 0.244) | $x_{37}$ | $A_{43}$ | STX | (-0.173, -0.027, 0.265, 0.411) |
| $x_{14}$ | $A_{12}$ | APT | (-0.176, -0.07, 0.142, 0.248) | $x_{38}$ | $A_{14}$ | CORE | (-0.291, -0.001, 0.579, 0.869) |
| $x_{15}$ | $A_{29}$ | NEAR | (-0.171, -0.036, 0.233, 0.368) | $x_{39}$ | $A_{42}$ | EGLD | (-0.194, -0.112, 0.052, 0.134) |
| $x_{16}$ | $A_{18}$ | ONDO | (-0.144, -0.012, 0.252, 0.384) | $x_{40}$ | $A_{34}$ | MANA | (-0.188, -0.045, 0.241, 0.384) |
| $x_{17}$ | $A_{23}$ | TON | (-0.15, -0.055, 0.135, 0.229) | $x_{41}$ | $A_{27}$ | AR | (-0.203, -0.024, 0.333, 0.512) |
| $x_{18}$ | $A_{25}$ | GNO | (-0.123, -0.04, 0.127, 0.21) | $x_{42}$ | $A_6$ | LINK | (-0.15, -0.032, 0.204, 0.322) |
| $x_{19}$ | $A_{33}$ | ONE | (-0.198, -0.084, 0.144, 0.258) | $x_{43}$ | $A_{22}$ | RUNE | (-0.182, -0.058, 0.191, 0.315) |
| $x_{20}$ | $A_{38}$ | LTC | (-0.184, -0.091, 0.094, 0.186) | $x_{44}$ | $A_{16}$ | MNT | (-0.101, 0.007, 0.222, 0.33) |
| $x_{21}$ | $A_{41}$ | FTM | (-0.189, -0.068, 0.174, 0.294) | $x_{45}$ | $A_{30}$ | 1INCH | (-0.235, -0.115, 0.125, 0.245) |
| $x_{22}$ | $A_{37}$ | VET | (-0.166, -0.032, 0.237, 0.372) | $x_{46}$ | $A_{19}$ | NUM | (-0.228, 0.116, 0.805, 1.15) |
| $x_{23}$ | $A_9$ | AVAX | (-0.166, -0.026, 0.253, 0.393) | $x_{47}$ | $A_{28}$ | DYDX | (-0.22, -0.087, 0.18, 0.314) |
| $x_{24}$ | $A_{26}$ | AAVE | (-0.172, -0.059, 0.169, 0.282) | | | | |

Source: Authors' own compilation

are detailed in Table 13. By analyzing these larger groups, we can better understand the dynamics of risk and return when more assets are considered. This broader perspective helps us assess the robustness of our model, particularly in terms of its adaptability and performance under different conditions.

As observed in the results, assets $A_3$ (TRX, rank 5) and $A_{15}$ (CRO, rank 9) were included in all portfolio constructions, regardless of the input data configurations and scenarios. In addition, assets UNI (rank 8) and $A_{17}$ were present in all scenarios and input data sets, with the exception of the portfolio that utilized 47 input data points and a cardinality of $k = 3$. Notably, all these assets belong to the top 10 category, classified as excellent assets. This consistent presence across various portfolios further emphasizes the reliability and quality of these selections in optimizing investment outcomes.

As reflected in the results, the model successfully constructed robust portfolios across different input data configurations. It effectively generated various scenarios tailored for different types of investors. These outcomes underscore the efficiency of our proposed model, demonstrating its capability to adapt to diverse requirements. Notably, the results show that portfolios formed from high-quality assets outperform those generated from other datasets. This finding highlights the critical importance of the preselection process prior to portfolio optimization, as selecting superior assets enhances overall portfolio performance. The analysis confirms that thoughtful asset selection is vital for achieving optimal investment outcomes.

**Table 12. Asset weight allocations for top 10 assets under different cardinality constraints.**

| No. | $k$ | Objective Function | $x_1$ A$_1$ (ETH) | $x_2$ A$_4$ (BNB) | $x_3$ A$_5$ (BTC) | $x_5$ A$_3$ (TRX) | $x_6$ A$_8$ (SUI) | $x_8$ A$_{15}$ (UNI) | $x_9$ A$_{17}$ (CRO) |
|---|---|---|---|---|---|---|---|---|---|
| Scenario 1 | $k = 3$ | -0.070 | – | – | – | 50% | – | 10% | 40% |
| Scenario 2 | $k = 5$ | -0.052 | – | 10% | – | 50% | 10% | 10% | 20% |
| Scenario 3 | $k = 7$ | -0.033 | 10% | 10% | 10% | 40% | 10% | 10% | 10% |

Source: Authors' own computation

**Table 13. Asset weight allocations for larger asset groups: top 20, 30, 40, and 47.**

**Input Data: Top 20 Assets ($x_1$ to $x_{20}$)**

| No. | $k$ | Objective Function | $x_2$ A$_4$ (BNB) | $x_5$ A$_3$ (TRX) | $x_6$ A$_8$ (SUI) | $x_8$ A$_{15}$ (UNI) | $x_9$ A$_{17}$ (CRO) | $x_{12}$ A$_{47}$ (ALGO) | $x_{16}$ A$_{18}$ (ONDO) |
|---|---|---|---|---|---|---|---|---|---|
| Scenario 1 | $k = 3$ | -0.070 | – | 50% | – | 10% | 40% | – | – |
| Scenario 2 | $k = 5$ | -0.054 | – | 50% | – | 10% | 20% | 10% | 10% |
| Scenario 3 | $k = 7$ | -0.036 | 10% | 40% | 10% | 10% | 10% | 10% | 10% |

**Input Data: Top 30 Assets ($x_1$ to $x_{30}$)**

| No. | $k$ | Objective Function | $x_5$ A$_3$ (TRX) | $x_8$ A$_{15}$ (UNI) | $x_9$ A$_{17}$ (CRO) | $x_{12}$ A$_{47}$ (ALGO) | $x_{16}$ A$_{18}$ (ONDO) | $x_{27}$ A$_{40}$ (CELO) | $x_{29}$ A$_{46}$ (XTZ) |
|---|---|---|---|---|---|---|---|---|---|
| Scenario 1 | $k = 3$ | -0.070 | 50% | 10% | 40% | – | – | – | – |
| Scenario 2 | $k = 5$ | -0.055 | 50% | 10% | 20% | – | 10% | – | 10% |
| Scenario 3 | $k = 7$ | -0.038 | 40% | 10% | 10% | 10% | 10% | 10% | 10% |

**Input Data: Top 40 Assets ($x_1$ to $x_{40}$)**

| No. | $k$ | Objective Function | $x_5$ A$_3$ (TRX) | $x_8$ A$_{15}$ (UNI) | $x_9$ A$_{17}$ (CRO) | $x_{16}$ A$_{18}$ (ONDO) | $x_{27}$ A$_{40}$ (CELO) | $x_{29}$ A$_{46}$ (XTZ) | $x_{38}$ A$_{14}$ (CORE) |
|---|---|---|---|---|---|---|---|---|---|
| Scenario 1 | $k = 3$ | -0.070 | 50% | 10% | 40% | – | – | – | – |
| Scenario 2 | $k = 5$ | -0.057 | 50% | 10% | 20% | 10% | – | – | 10% |
| Scenario 3 | $k = 7$ | -0.041 | 40% | 10% | 10% | 10% | 10% | 10% | 10% |

**Input Data: All Assets ($x_1$ to $x_{47}$)**

| No. | $k$ | Objective Function | $x_5$ A$_3$ (TRX) | $x_8$ A$_{15}$ (UNI) | $x_9$ A$_{17}$ (CRO) | $x_{16}$ A$_{18}$ (ONDO) | $x_{38}$ A$_{14}$ (CORE) | $x_{44}$ A$_{16}$ (MNT) | $x_{46}$ A$_{19}$ (NUM) |
|---|---|---|---|---|---|---|---|---|---|
| Scenario 1 | $k = 3$ | -0.105 | 40% | – | 10% | – | – | – | 50% |
| Scenario 2 | $k = 5$ | -0.094 | 20% | 10% | 10% | – | 10% | – | 50% |
| Scenario 3 | $k = 7$ | -0.074 | 10% | 10% | 10% | 10% | 10% | 10% | 40% |

Source: Authors' own computation

## 5. Discussion and Practical implications

This paper introduces a two-stage framework designed to enhance cryptocurrency portfolio performance, addressing a critical gap in the literature and providing practical tools for investors.

The first stage emphasizes the pre-selection of high-potential assets through an innovative approach grounded in MADM methods. This strategy enables investors to systematically identify and evaluate assets that exhibit promising characteristics, significantly enhancing the asset selection process. Given the inherent volatility and unpredictability of the cryptocurrency market, the emphasis on high-quality pre-selection highlights the crucial need for rigorous asset evaluation. Historically, the lack of comprehensive studies addressing asset pre-selection in cryptocurrency portfolio optimization has resulted in the absence of established performance criteria specifically tailored for evaluating cryptocurrencies. This paper addresses this critical gap by introducing comprehensive criteria for the pre-selection of cryptocurrency assets. By providing a structured methodology, the research empowers investors to cherry-pick high-quality assets based on systematic evaluations rather than relying on intuition or superficial analysis. From a methodological perspective, this paper also contributes a robust framework for ranking and pre-selecting crypto-assets grounded in MADM techniques. One of the notable challenges associated with MADM approaches is that different methods can yield varying results, creating uncertainty regarding which technique provides the most reliable outcomes. To combat this issue, this paper presents a unified framework for pre-selection that synthesizes results from multiple MADM methods. By combining insights from various approaches, the framework enhances reliability and fosters greater trust in the asset selection process. Moreover, the implications of this framework extend beyond cryptocurrency. Its methodologies can be applied in various fields within decision science and engineering, where the ranking and ordering of alternatives are essential. This versatility underscores the framework's potential to inform asset selection processes across different asset classes and investment contexts, ultimately leading to more effective decision-making. Therefore, the first stage of this two-stage framework not only fills a significant gap in the literature but also provides practical tools and methodologies that empower investors to make informed decisions in the rapidly evolving cryptocurrency landscape.

In the second stage, the focus shifts to the optimization process itself. Recognizing that uncertainty is a fundamental characteristic of capital markets, the framework incorporates credibility theory within the CVaR framework. By integrating these methodologies, the model effectively leverages their combined strengths to address downside risk and manage uncertainty during the optimization phase. This dual approach not only enhances the robustness of the portfolio but also empowers investors to make informed decisions in a volatile market landscape. Integrating credibility theory with traditional risk management frameworks equips managers with the tools necessary to navigate the complexities of cryptocurrency investments. This framework enables decision-makers to assess and account for uncertainties, leading to more resilient investment strategies. The ability to model downside risk effectively aids in capital preservation, particularly in a market characterized by rapid fluctuations and unpredictable events.

Therefore, the two-stage framework presented in this paper offers a structured and comprehensive pathway for enhancing cryptocurrency portfolio performance. By focusing on high-quality pre-selection in the first stage and sophisticated optimization in the second, this framework provides invaluable insights for investors seeking to navigate the intricacies of the cryptocurrency market. This holistic approach not only aids in optimizing investment strategies but also prepares investors to respond effectively to the challenges posed by market volatility and uncertainty. Furthermore, the methodologies introduced in this paper have broad applicability beyond cryptocurrency, providing valuable insights for asset selection processes across various investment domains. The principles of the proposed two-stage framework can be seamlessly applied to traditional financial markets, such as stocks, bonds, and alternative investments, thereby enhancing the asset selection process and optimizing portfolio construction in these markets as well. By integrating systematic pre-selection with advanced optimization techniques, this framework equips investors and managers with powerful tools for better decision-making, ensuring that portfolios are aligned with both risk and return objectives.

While the two-stage framework presented in this paper offers a structured and innovative approach to cryptocurrency portfolio optimization, several limitations should be acknowledged. First, the methodology relies on the availability of high-quality and consistent data, which can be a significant challenge in the cryptocurrency market due to its rapid evolution and volatility. Data gaps, especially in newly launched cryptocurrencies, may affect the accuracy of the asset pre-selection process. Additionally, while the integration of credibility theory with CVaR improves risk management, it may not account for all potential market dynamics, particularly during extreme market events. Furthermore, the framework also assumes that investors have access to sufficient computational resources and expertise to implement the methodologies effectively, which may limit its applicability to some investors. Lastly, while the proposed methods have been demonstrated in the context of cryptocurrency, further research is needed to test their effectiveness across other asset classes and market conditions. These limitations suggest that while the framework provides valuable insights, its real-world application may require adaptation to specific contexts and additional refinements.

## 6. Conclusions

This study presents a comprehensive two-stage framework aimed at enhancing cryptocurrency portfolio performance, specifically designed to tackle the challenges of portfolio construction in volatile and uncertain cryptocurrency markets. The framework comprises two sequential stages: Stage 1 focuses on the pre-selection of high-potential assets using a novel asset pre-selection approach. This process begins with the evaluation of 47 cryptocurrency assets, which are sorted based on comprehensive performance criteria. The assets are then categorized into five distinct groups, allowing for a systematic assessment. From these groups, we identify the top-performing segment, selecting the top 10 assets that exhibit the highest potential for returns. Stage 2 optimizes the selected assets through the application of a credibilistic CVaR model, while also considering various cardinality constraints to construct different scenarios for portfolio allocation. The effectiveness of this two-stage framework was rigorously tested by comparing the resultant portfolios against those constructed from the other asset groups. The results of this two-stage framework demonstrate its effectiveness in constructing well-diversified and efficient portfolios, addressing both the challenges of asset pre-selection and the complexities associated with uncertainty. By integrating these methodologies, investors are better equipped to navigate the risks associated with cryptocurrency investments while maximizing potential returns. This innovative approach not only enhances portfolio performance but also provides valuable insights for investors operating in the dynamic landscape of cryptocurrency markets.

Although the proposed framework represents a significant advancement in enhancing cryptocurrency portfolio performance, several avenues for further research remain. One promising area is the development of robust performance criteria that can be applied to other markets, such as equities, bonds, and commodities. This would facilitate the implementation of the pre-selection process beyond cryptocurrencies, enabling a broader application of the framework. Additionally, exploring alternative fuzzy number representations, such as coherent fuzzy numbers, could provide a more nuanced understanding of investor expectations. Traditional fuzzy numbers may not adequately represent the complexities of investor sentiment, especially in volatile markets. By investigating how these alternative representations influence portfolio performance and risk management, researchers could uncover new insights that enhance decision-making processes. Furthermore, the reliability of expert opinions in gathering fuzzy data is another critical aspect to consider. Expert insights are often essential in shaping investment decisions, particularly in uncertain environments like cryptocurrency markets. However, the subjective nature of these opinions can introduce bias. By utilizing Z-number theory, researchers could develop a more structured framework for incorporating expert insights, mitigating potential biases while enhancing the credibility of the data. This approach could result in more accurate models that reflect true market conditions, ultimately improving portfolio outcomes.

It is crucial to acknowledge that digital assets, especially cryptocurrencies, are continually evolving and are marked by substantial volatility and inherent risks. The value of most cryptocurrencies is largely influenced by speculative trading,

where market sentiment is a key driver. Therefore, investors must exercise caution and perform comprehensive due diligence before making investment decisions in the cryptocurrency market, fully understanding its speculative nature and the various factors that can impact market dynamics.

## Supporting information

**S1 Appendix. Appendix S1: Pseudo Code of Selected MADM Algorithms and Required Further Calculations.** (DOCX)

## Author contributions

**Conceptualization:** Hossein Ghanbari, Ronald Ravinesh Kumar.

**Data curation:** Hossein Ghanbari, Sina Tavakoli, Mostafa Shabani, Ronald Ravinesh Kumar.

**Formal analysis:** Hossein Ghanbari, Sina Tavakoli, Mostafa Shabani.

**Investigation:** Hossein Ghanbari.

**Methodology:** Hossein Ghanbari, Sina Tavakoli, Mostafa Shabani.

**Project administration:** Hossein Ghanbari.

**Resources:** Hossein Ghanbari.

**Software:** Hossein Ghanbari.

**Supervision:** Emran Mohammadi, Seyed Jafar Sadjadi, Ronald Ravinesh Kumar.

**Validation:** Hossein Ghanbari, Emran Mohammadi.

**Visualization:** Hossein Ghanbari.

**Writing – original draft:** Hossein Ghanbari.

**Writing – review & editing:** Hossein Ghanbari, Emran Mohammadi, Seyed Jafar Sadjadi, Ronald Ravinesh Kumar.

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
