## [Decision Letter · Decision Letter 0]

Dear Dr. Mohammadi,

Thank you for submitting your manuscript to PLOS ONE. After careful consideration, we feel that it has merit but does not fully meet PLOS ONE’s publication criteria as it currently stands. Therefore, we invite you to submit a revised version of the manuscript that addresses the points raised during the review process.

**ACADEMIC EDITOR: please revise accordingly**

We look forward to receiving your revised manuscript.

Kind regards,

Zhengmao Li

Academic Editor

PLOS ONE

Reviewers' comments:

Reviewer's Responses to Questions

**Comments to the Author**

1. Is the manuscript technically sound, and do the data support the conclusions?

Reviewer #1: Yes

Reviewer #2: Yes

2. Has the statistical analysis been performed appropriately and rigorously?

Reviewer #1: Yes

Reviewer #2: Yes

3. Have the authors made all data underlying the findings in their manuscript fully available?

Reviewer #1: Yes

Reviewer #2: Yes

4. Is the manuscript presented in an intelligible fashion and written in standard English?

Reviewer #1: Yes

Reviewer #2: Yes

Reviewer #1: I have carefully reviewed this manuscript; the following are my comments and suggestions.

Comments and Suggestions

1. There are two recent studies that substantially extend the extent literature on cryptocurrencies. Ali, Khurram, and Sensoy (2025) indicate the importance of cryptocurrencies in portfolio optimization (risk reduction), whereas Ali, Du, and Majeed (2025) indicate the importance of matching trading periods while estimating intermarket interconnectedness. While the former will help the authors to improve the first paragraph of the Introduction Section, the latter will help the authors to improve the second paragraph. https://doi.org/10.1186/s40854-024-00686-4; https://doi.org/10.1016/j.frl.2025.107016

2. The authors have discussed dome seminal studies that have employed VaR and CVaR; however, it is important to mention studies that have recently employed these methods. In the wake of consistently evolving econometric tools to understand diversification, it is fundamental to discuss recent high-quality work that has used this method. I suggest taking guidance from the following study to strengthen the use of the VaR and CVaR methods (https://doi.org/10.1016/j.rser.2023.114137 & https://doi.org/10.1016/j.ribaf.2022.101768 ).

3. The above four studies will also help the authors to improve the Literature Review section of the study.

4. Sections 3 and 4 occupy nearly 80% of the manuscript size, indicating sections are not well-distributed.

5. Please reduce the size of the study; it is very lengthy and hard to keep interest while reading it.

6. The authors need to discuss the limitations of the study.

7. Finally, discuss some practical implications of the study.

Overall, once my comments are addressed, the manuscript can be reconsidered for publication. Good luck!

Reviewer #2: This paper proposes a two - stage framework to enhance the performance of cryptocurrency portfolios. The research topic is of practical significance, and the research methods are relatively systematic. My comments are as follows:

It is advisable to conduct comparative experiments by comparing with other methods applicable to cryptocurrency asset pre - selection, so as to highlight the advantages and innovativeness of the method proposed in this paper.

In the dataset, the data of the asset ONDO only covers 330 days, which is different from the 379 - day data of other assets. This may seriously affect the reliability of statistical analysis and the universality of conclusions.

When constructing the Credibilistic CVaR model, the trapezoidal fuzzy variables are obtained based on expert opinions. It is necessary to consider combining market data and objective algorithms to determine the fuzzy variables, or adding multiple sets of expert opinions for comparative analysis to reduce subjectivity and enhance the objectivity and accuracy of the model.

When presenting the results of asset pre - selection and portfolio optimization, the economic significance and practical application value of the results are insufficiently elaborated.

In addition to the mentioned uncertainties and market volatility, the significant impacts of regulatory policy changes, technological innovations, and other factors on the cryptocurrency market should also be considered.

Although a review of relevant literature has been carried out, the exploration in some fields is not in - depth enough. It is necessary to further analyze the limitations of existing research more deeply and clearly explain how this research can address these issues specifically, highlighting the necessity and innovativeness of the research.

It is recommended to supplement the following literature (not mine) 10.1109/TCSI.2024.3523339

10.1109/TII.2024.3495785

**Do you want your identity to be public for this peer review?** For information about this choice, including consent withdrawal, please see our Privacy Policy

Reviewer #1: **Yes: ** Fahad Ali

Reviewer #2: No

---

## [Author Response · Author response to Decision Letter 1]

7 Apr 2025

COVER LETTER FOR RESUBMISSION OF MANUSCRIPT

Dear Editor

Prof. Dr. Emily Chenette

We are enclosing herewith a revised version of our manuscript entitled “A Two-Stage Framework for Enhancing Cryptocurrency Portfolio Performance: Integrating Credibilistic CVaR Criterion with a Novel Asset Preselection Approach” for possible publication in PLOS One.

The reviewer comments were highly insightful and enabled us to considerably enhance the quality of our manuscript. In the following pages are our point-by-point responses to each of the comments of the reviewer. Moreover, new discussions and modifications in some formulas and tables have been done to eliminate the previous document's ambiguities and inaccuracies.

We shall look forward to hearing from you at your earliest convenience.

Sincerely yours,

Emran Mohammadi, Associate Professor

Title of the paper: A Two-Stage Framework for Enhancing Cryptocurrency Portfolio Performance: Integrating Credibilistic CVaR Criterion with a Novel Asset Preselection Approach

First of all, the authors would like to thank all the anonymous reviewers and the Associate Editor again for their efforts and valuable time to review and improve our paper. Taking into account their constructive suggestions and comments, the paper has been carefully revised following the referees’ comments.

We provide below the responses to each referee's comments. All amends and changes in the revised version are yellow highlighted.

---

## [Decision Letter · Decision Letter 1]

A Two-Stage Framework for Enhancing Cryptocurrency Portfolio Performance: Integrating Credibilistic CVaR Criterion with a Novel Asset Preselection Approach

PONE-D-25-06239R1

Dear Dr. Mohammadi,

We’re pleased to inform you that your manuscript has been judged scientifically suitable for publication and will be formally accepted for publication once it meets all outstanding technical requirements.

Kind regards,

Zhengmao Li

Academic Editor

PLOS ONE

Additional Editor Comments (optional):

Reviewers' comments:

Reviewer's Responses to Questions

**Comments to the Author**

Reviewer #1: All comments have been addressed

Reviewer #2: All comments have been addressed

2. Is the manuscript technically sound, and do the data support the conclusions?

Reviewer #1: Yes

Reviewer #2: (No Response)

3. Has the statistical analysis been performed appropriately and rigorously?

Reviewer #1: Yes

Reviewer #2: (No Response)

4. Have the authors made all data underlying the findings in their manuscript fully available?

Reviewer #1: Yes

Reviewer #2: (No Response)

5. Is the manuscript presented in an intelligible fashion and written in standard English?

Reviewer #1: Yes

Reviewer #2: (No Response)

Reviewer #1: The authors have addressed my comments, and the manuscript is substantially improved. However, I suggest carefully proofreading and reducing the size of the abstract. Besides this, I have no more comments, and it can be considered for publication. Good luck!

Reviewer #2: (No Response)

**Do you want your identity to be public for this peer review?** For information about this choice, including consent withdrawal, please see our Privacy Policy

Reviewer #1: No

Reviewer #2: No

---

## [Editor Report · Acceptance letter]

PONE-D-25-06239R1

PLOS ONE

Dear Dr. Mohammadi,

I'm pleased to inform you that your manuscript has been deemed suitable for publication in PLOS ONE. Congratulations! Your manuscript is now being handed over to our production team.

Kind regards,

on behalf of

Dr Zhengmao Li

Academic Editor

PLOS ONE